# Scalable Interpretability via Polynomials

**Abhimanyu Dubey**
Meta AI
dubeya@fb.com

**Filip Radenovic**
Meta AI
filipradenovic@fb.com

**Dhruv Mahajan**
Meta AI
dhruvm@fb.com

## Abstract

Generalized Additive Models (GAMs) have quickly become the leading choice for interpretable machine learning. However, unlike uninterpretable methods such as DNNs, they lack expressive power and easy scalability, and are hence not a feasible alternative for real-world tasks. We present a new class of GAMs that use tensor rank decompositions of polynomials to learn powerful, *inherently-interpretable* models. Our approach, titled Scalable Polynomial Additive Models (SPAM) is effortlessly scalable and models *all* higher-order feature interactions without a combinatorial parameter explosion. SPAM outperforms all current interpretable approaches, and matches DNN/XGBoost performance on a series of real-world benchmarks with up to hundreds of thousands of features. We demonstrate by human subject evaluations that SPAMs are demonstrably more interpretable in practice, and are hence an effortless replacement for DNNs for creating interpretable and high-performance systems suitable for large-scale machine learning. Source code is available at github.com/facebookresearch/nbm-spam.

## 1 Introduction

Interpretable machine learning systems suffer from a tradeoff between approximation and interpretability: high-performing models used in practice have millions of parameters and are highly non-linear, making them difficult to interpret. *Post-hoc* explainability attempts to remove this tradeoff by explaining black-box predictions with interpretable instance-specific approximations (e.g., LIME [Ribeiro et al., 2016], SHAP [Lundberg and Lee, 2017]), however, they are notoriously unstable [Ghorbani et al., 2019a, Lakkaraju and Bastani, 2020], expensive to compute [Slack et al., 2021], and in many cases, inaccurate [Lipton, 2018]. It is therefore desirable to learn models that are instead *inherently-interpretable* (glass-box), i.e., do not require *post-hoc* interpretability, but provide better performance than *inherently-interpretable* classical approaches such as linear or logistic regression.

This has led to a flurry of research interest in Generalized Additive Models [Hastie and Tibshirani, 2017], i.e., methods that non-linearly transform each input feature separately, e.g., tree-based GAM and GA$^2$Ms [Lou et al., 2013], or NAMs [Agarwal et al., 2021]. While these are an improvement over linear methods, due to their highly complex training procedures, they are not a simple drop-in replacement for black-box methods. Furthermore, it is not straightforward to model feature interactions in these models: the combinatorial explosion in higher-order interactions makes this infeasible without a computationally expensive feature selection [Lou et al., 2013, Chang et al., 2021].

Contrary to these approaches, we revisit an entirely different solution: *polynomials*. It is folklore in machine learning that since polynomials are universal approximators [Stone, 1948, Weierstrass, 1885], hypothetically, modeling all possible feature interactions will suffice in creating powerful learners and eliminate the need for intricate non-linear transformations of data. Furthermore, polynomials can also provide a rubric for interpretability: lower-degree polynomials, that have interactions between a few features are interpretable but likely less accurate, whereas polynomials of higher degree have larger predictive power at the cost of interpretability. Thus, not only are polynomial models capable of capturing all possible interactions between features, they also give practitioners the ability to select a suitable model (order of interactions, i.e., degree) based precisely on their requirement in the

36th Conference on Neural Information Processing Systems (NeurIPS 2022).

interpretability-performance tradeoff. However, such an approach has till date remained elusive, as learning interpretable feature interactions *efficiently and at scale* remains an open problem.

In this paper, we introduce a highly scalable approach to modeling feature interactions for inherently interpretable classifiers based on *rank decomposed polynomials*. Our contributions are:

1. First, we present an algorithm titled Scalable Polynomial Additive Models (**SPAM**) to learn *inherently-interpretable* classifiers that can learn all possible feature interactions by leveraging *low-rank tensor decompositions* of polynomials [Nie, 2017b]. To the best of our knowledge, **SPAM** is the among the first Generalized Additive Models with feature interactions scalable to 100K features. SPAM can be trained end-to-end with SGD (backpropagation) and GPU acceleration, but has orders of magnitude fewer parameters than comparable interpretable models.
2. We demonstrate that under a coarse regularity assumption (Assumption 1), SPAM converges to the optimal polynomial as the number of samples $n \to \infty$. Furthermore, we establish novel non-asymptotic excess risk bounds that match classic bounds for linear or full-rank polynomials.
3. SPAM outperforms all current interpretable baselines on several machine learning tasks. To the best of our knowledge, our experimental benchmark considers problems of size (number of samples, and dimensionality) orders of magnitude larger than prior work, and we show that our simple approach scales easily to all problems, whereas prior approaches fail. Moreover, we demonstrate that the need for modeling feature interactions is as important as non-linear feature transformations: on most real-world tabular datasets, *pairwise* interactions suffice to match DNN performance.
4. We conduct a detailed human subject evaluation to corroborate our performance with practical interpretability. We demonstrate that SPAM explanations are more faithful compared to *post-hoc* methods, making them a *drop-in replacement for black-box methods* such as DNNs for interpretable machine learning on tabular and concept-bottleneck [Koh et al., 2020] problems.

## 2 Related Work

Here we survey the literature most relevant to our work, i.e., interpretable ML with feature interactions, and polynomials for machine learning. Please see Appendix Section B for a detailed survey.

**Transparent and Interpretable Machine Learning**. Early work has focused on greedy or ensemble approaches to modeling interactions [Friedman, 2001, Friedman and Popescu, 2008] that enumerate pairwise interactions and learn additive interaction effects. Such approaches often pick spurious interactions when data is sparse [Lou et al., 2013] and are impossible to scale to modern-sized datasets due to enumeration of individual combinations. As an improvement, Lou et al. [2013] proposed GA$^2$M that uses a statistical test to select only "true" interactions. However GA$^2$M fails to scale to large datasets as it requires constant re-training of the model and ad-hoc operations such as discretization of features which may require tuning for datasets with a large dimensionality. Other generalized additive models require expensive training of millions of decision trees, kernel machines or splines [Hastie and Tibshirani, 2017], making them unattractive compared to black-box models.

An alternate approach is is to learn interpretable neural network transformations. Neural Additive Models (NAMs, Agarwal et al. [2021]) learn a DNN per feature. TabNet [Arık and Pfister, 2021] and NIT [Tsang et al., 2018] alternatively modify NN architectures to increase their interpretability. NODE-GAM [Chang et al., 2021] improves NAMs with oblivious decision trees for better performance while maintaining interpretability. Our approach is notably distinct from these prior works: we do not require iterative re-training; we can learn *all* pairwise interactions regardless of dimensionality; we can train SPAM via backpropagation; and we scale effortlessly to very large-scale datasets.

**Learning Polynomials**. The idea of decomposing polynomials was of interest prior to the deep learning era. Specifically, the work of Ivakhnenko [1971], Oh et al. [2003], Shin and Ghosh [1991] study learning neural networks with polynomial interactions, also known as *ridge polynomial networks* (RPNs). However, RPNs are typically not interpretable: they learn interactions of a very high order, and include non-linear transformation. Similar rank decompositions have been studied in the context of matrix completion [Recht, 2011], and are also a subject of interest in tensor decompositions [Nie, 2017a, Brachat et al., 2010], where, contrary to our work, the objective is to decompose existing tensors rather than directly learn decompositions from gradient descent. Recently, Chrysos et al. [2019, 2020] use tensor decompositions to learn higher-order polynomial relationships in intermediate layers of generative models. However, their work uses a recursive formulation and learns high-degree polynomials directly from uninterpretable input data (e.g., images), and hence is non-interpretable.

# 3 Scalable Polynomial Additive Models

**Setup and Notation**. Matrices are represented by uppercase boldface letters, e.g., $\mathbf{X}$ and vectors by boldface lowercase, i.e., $\boldsymbol{x}$. We assume that the covariates lie within the set $\mathcal{X} \subseteq \mathbb{R}^d$, and the labels lie within the finite set $\mathcal{Y}$. Data $(\boldsymbol{x}, y) \in \mathcal{X} \times \mathcal{Y}$ are drawn following some unknown (but fixed) distribution $\mathfrak{P}$. We assume we are provided with $n$ i.i.d. samples $\{(\boldsymbol{x}_i, y_i)\}_{i=1}^n$ as the training set.

**Motivation**. Generalized Additive Models [Hastie and Tibshirani, 2017] are an excellent design choice for interpretable models, as they learn transformations of individual features, allowing us to model exactly the contribution of any feature. A typical GAM is as follows.

$$f(\boldsymbol{x}) = \sum_{i=1}^d \underbrace{f_i(x_i)}_{\text{order 1 (unary)}} + \sum_{j>i}^d \underbrace{f_{ij}(x_i, x_j)}_{\text{order 2 (pairwise)}} + \cdots + \underbrace{f_{1\ldots d}(\boldsymbol{x})}_{\text{order } d}.$$

(Interpretability $\longleftarrow$ ; Performance $\longrightarrow$)

Where $f_i$, etc., are possibly non-linear transformations. It is evident [Lou et al., 2013] that as the order of interaction increases, e.g., beyond pairwise interactions, these models are no longer interpretable, albeit at some benefit to performance. While some approaches (e.g., Agarwal et al. [2021], Chang et al. [2021]) learn $f_i$ via neural networks, we want to learn the simplest GAMs, i.e., polynomials. Specifically, we want to learn a polynomial $P(\boldsymbol{x})$, of order (degree) $k \leq d$ of the form:

$$P(\boldsymbol{x}) = b + \sum_{i=1}^d w_i^{(1)} \cdot x_i + \sum_{i,j}^d w_{ij}^{(2)} \cdot x_i x_j + \ldots \sum_{i_1 \ldots, i_k}^d \left( w_{i_1 \ldots i_k}^{(k)} \cdot \prod_{j=1}^k x_{i_j} \right).$$

Here, the weights $\mathbf{W}^{(l)} = \{w_{i_1 \ldots i_l}^{(l)}\}, 1 \leq l \leq k$ capture the $l-$ary interactions between subsets of $l$ features. For small values of $d$ and $k$, one can potentially enumerate all possible interactions and learn a linear model of $\mathcal{O}(d^k)$ dimensionality, however this approach quickly becomes infeasible for large $d$ and $k$. Furthermore, the large number of parameters in $P(\boldsymbol{x})$ make regularization essential for performance, and the computation of each interaction can be expensive. Alternatively, observe that any polynomial that models $k-$ary interactions can be written as follows, for weights $\{\mathbf{W}^{(l)}\}_{l=1}^k$,

$$P(\boldsymbol{x}) = \mathbf{W}^{(1)} \odot_1 \boldsymbol{x} + \mathbf{W}^{(2)} \odot_2 \boldsymbol{x} + \ldots \mathbf{W}^{(k)} \odot_k \boldsymbol{x} + b.$$

Here, the weights $\mathbf{W}^{(l)} \in \mathbb{R}^{d^l}$ are written as a tensor of order $l$, and the operation $\odot_l$ refers to a repeated inner product along $l$ dimensions, e.g., $\mathbf{W} \in \mathbb{R}^{d^2}$, $\boldsymbol{x}^\top \mathbf{W} \boldsymbol{x} = \mathbf{W} \odot_2 \boldsymbol{x} = (\mathbf{W} \odot \boldsymbol{x}) \odot \boldsymbol{x}$ where $\odot$ denotes the inner product. More generally, any *symmetric* order $l$ tensor admits an equivalent polynomial representation having *only* degree$-l$ terms. Now, this representation is still plagued with the earlier curse of dimensionality for arbitrary weight tensors $\{\mathbf{W}^{(l)}\}_{l=1}^k$, and we will now demonstrate how to circumvent this issue by exploiting rank decompositions.

## 3.1 Learning Low-Rank Decompositions of Polynomials

The primary insight of our approach is to observe that any *symmetric* tensor $\mathbf{W}^{(l)}$ of order $l$ also admits an additional *rank decomposition* [Nie, 2017a, Brachat et al., 2010] ($\otimes$ being the outer product):

$$\mathbf{W}^{(l)} = \sum_{i=1}^r \lambda_i \cdot \underbrace{\boldsymbol{u}_i \otimes \boldsymbol{u}_i \cdots \otimes \boldsymbol{u}_i}_{l \text{ times}}.$$

Where $\{\boldsymbol{u}_i\}_{i=1}^r$ are the (possibly orthonormal) $d$ dimensional basis vectors, $r \in \mathbb{Z}_+$ denotes the *rank* of the tensor, and the scalars $\{\lambda_i\}_{i=1}^r, \lambda_i \in \mathbb{R}$ denote the singular values. Our objective is to directly learn the rank decomposition of an order $l$ tensor, and therefore learn the required polynomial function. This gives us a total of $\mathcal{O}(rd)$ learnable parameters per tensor, down from a previous dependence of $\mathcal{O}(d^l)$. This *low-rank* formulation additionally enables us to compute the polynomial more efficiently. Specifically, for a degree $k$ polynomial with ranks $\boldsymbol{r} = \{1, r_2, \ldots, r_k\}$, we have:

$$P(\boldsymbol{x}) = b + \langle \boldsymbol{u}_1, \boldsymbol{x} \rangle + \sum_{i=1}^{r_2} \lambda_{2i} \cdot \langle \boldsymbol{u}_{2i}, \boldsymbol{x} \rangle^2 + \sum_{i=1}^{r_3} \lambda_{3i} \cdot \langle \boldsymbol{u}_{3i}, \boldsymbol{x} \rangle^3 \ldots + \sum_{i=1}^{r_k} \lambda_{ki} \cdot \langle \boldsymbol{u}_{ki}, \boldsymbol{x} \rangle^k. \quad (1)$$

Where $\{\boldsymbol{u}_{li}\}_{i=1}^{r_l}, \{\lambda_{li}\}_{i=1}^{r_l}$ denote the corresponding bases and eigenvalues for $\mathbf{W}^{(l)}$. The above term can now be easily computed via a simple inner product, and has a time as well as space complexity

of $\mathcal{O}(d\|\boldsymbol{r}\|_1)$, instead of the earlier $\mathcal{O}(d^k)$. Here, $\|\boldsymbol{r}\|_1$ simply represents the sum of ranks of all the $k$ tensors. Note that *low-rank* decompositions do not enjoy a free lunch: the rank $r$ of each weight tensor can, in the worst case, be polynomial in $d$. However, it is reasonable to assume that correlations in the data $\mathcal{X}$ will ensure that a small $r$ will suffice for real-world problems (see, e.g., Assumption 1).

**Optimization**. For notational simplicity, let us parameterize any order $k$ polynomial by the weight set $\boldsymbol{\theta} = \{b, \boldsymbol{u}_1, \{\lambda_{2i}, \boldsymbol{u}_{2i}\}_{i=1}^{r_2}, ..., \{\lambda_{ki}, \boldsymbol{u}_{ki}\}_{i=1}^{r_k}\} \in \mathbb{R}^{r(d+1)}$ where from now onwards $r = \|\boldsymbol{r}\|_1 = 1 + r_2 + r_3 + ... + r_k$ denotes the cumulative rank across all the tensors. We write the polynomial parametrized by $\boldsymbol{\theta}$ as $P(\cdot; \boldsymbol{\theta})$, and the learning problem for any loss $\ell$ is:

$$\text{select } \boldsymbol{\theta}_\star = \arg\min_{\boldsymbol{\theta} \in \boldsymbol{\Theta}} \sum_{i=1}^{n} \ell(P(\boldsymbol{x}_i; \boldsymbol{\theta}), y_i) + \beta \cdot \mathcal{R}(\boldsymbol{\theta}). \tag{2}$$

Here, $\boldsymbol{\Theta}$ denotes the feasible set for $\boldsymbol{\theta}$, and $\mathcal{R}(\cdot)$ denotes an appropriate regularization term scaled by $\beta > 0$. We can show that the above problem is well-behaved for certain data distributions $\mathcal{X}$.

**Proposition 1.** *If the regularization $\mathcal{R}$ and loss function $\ell : \mathcal{Y} \times \mathcal{Y} \rightarrow [0,1]$ are convex, $\lambda > 0$ and $\mathcal{X} \subset \mathbb{R}_+^d$ then the optimization problem in Equation 2 is convex in $\boldsymbol{\theta}$ for $\boldsymbol{\Theta} \subset \mathbb{R}_+^{r(d+1)}$.*

Proposition 1 suggests that if the training data is positive (achieved by renormalization), typical loss functions (e.g., cross-entropy or mean squared error) and $\mathcal{R}$ such as $L_p$ norms are well-suited for optimization. To ensure convergence, one can use proximal SGD with a small learning rate [Nitanda, 2014]. In practice, we find that unconstrained SGD also provides comparable performance.

### 3.2   Improving Polynomials for Learning

**Geometric rescaling**. Learning higher-order interactions is tricky as the order of interactions increases: the product of two features is an order of magnitude different from the original, and consequently, higher-order products are progressively disproportionate. To mitigate this, we rescale features such that (a) the scale is preserved across terms, and (b) the variance in interactions is captured. We replace the input $\boldsymbol{x}$ with $\tilde{\boldsymbol{x}}_l = \text{sign}(\boldsymbol{x}) \cdot |\boldsymbol{x}|^{1/l}$ for an interaction of order $l$. Specifically,

$$P(\boldsymbol{x}) = \langle \boldsymbol{u}_1, \boldsymbol{x} \rangle + \sum_{i=1}^{r_2} \lambda_{2i} \cdot \langle \boldsymbol{u}_{2i}, \tilde{\boldsymbol{x}}_2 \rangle^2 + \sum_{i=1}^{r_3} \lambda_{3i} \cdot \langle \boldsymbol{u}_{3i}, \tilde{\boldsymbol{x}}_3 \rangle^3 ... + \sum_{i=1}^{r_k} \lambda_{ki} \cdot \langle \boldsymbol{u}_{ki}, \tilde{\boldsymbol{x}}_k \rangle^k + b. \tag{3}$$

We denote this model as SPAM-LINEAR. We argue that this rescaling for unit-bounded features ensures higher interpretability as well as better learning (since all features are of similar variance). Note that for order 1, $\tilde{\boldsymbol{x}}_1 = \boldsymbol{x}$. For $k \geq 2$, consider a pairwise interaction between two features $x_i = 0.5$ and $x_j = 0.6$. Naively multiplying the two will give a feature value of $x_i x_j = 0.3$ (smaller than both $x_i$ and $x_j$), whereas an intuitive value of the feature would be $\sqrt{x_i x_j} = 0.54$, i.e., the geometric mean. Regarding the reduced variance, observe that unconstrained, the variance $\mathbb{V}(x_i x_j) \leq \mathbb{V}(x_i) \cdot \mathbb{V}(x_j)$ and $\mathbb{V}(\sqrt{x_i x_j}) \leq \mathbb{V}(\sqrt{x_i})\mathbb{V}(\sqrt{x_j}) \leq \sqrt{\mathbb{V}(x_i) \cdot \mathbb{V}(x_j)}$. If the features have small variance, e.g., $\mathbb{V}(x_i) = \mathbb{V}(x_j) = 10^{-2}$ and are uncorrelated, then the first case would provide a much smaller variance for the interaction, whereas the rescaling would preserve variance.

**Shared bases for multi-class problems**. When we are learning a multi-class classifier, it is observed that learning a unique polynomial for each class creates a large model (with $\mathcal{O}(2drC)$ weights) when the number of classes $C$ is large, which leads to overfitting and issues with regularization. Instead, we propose *sharing* bases ($\boldsymbol{u}$) across classes for all higher order terms, and learning *class-specific* singular values ($\lambda$) for all higher order terms ($\geq 2$) per class. For any input $\boldsymbol{x}$, $P(\boldsymbol{x}; \boldsymbol{\theta}) = \text{SOFTMAX}\{P_c(\boldsymbol{x}; \boldsymbol{\theta})\}_{c=1}^{C}$, where, for any class $c \in \{1, ..., C\}$,

$$P_c(\boldsymbol{x}; \boldsymbol{\theta}) = b_c + \langle \boldsymbol{u}_1^c, \boldsymbol{x} \rangle + \sum_{i=1}^{r_2} \lambda_{2i}^c \cdot \langle \boldsymbol{u}_{2i}, \tilde{\boldsymbol{x}}_2 \rangle^2 + \sum_{i=1}^{r_3} \lambda_{3i}^c \cdot \langle \boldsymbol{u}_{3i}, \tilde{\boldsymbol{x}}_3 \rangle^3 ... + \sum_{i=1}^{r_k} \lambda_{ki}^c \cdot \langle \boldsymbol{u}_{ki}, \tilde{\boldsymbol{x}}_k \rangle^k.$$

The terms in *green* denote weights unique to each class, and terms in *red* denote weights shared across classes. This reduces the model size to $\mathcal{O}((d+r)C + rd)$ from $\mathcal{O}(rdC)$. We set $\ell$ as the cross-entropy of the softmax of $\{P_c(\cdot, \boldsymbol{\theta})\}_{c=1}^{C}$, as both these operations preserve convexity.

**Exploring nonlinear input transformations**. Motivated by non-linear GAMs (e.g., GA$^2$M [Lou et al., 2013] and NAM [Agarwal et al., 2021]), we modify our low-rank decomposed algorithm by replacing $\tilde{\boldsymbol{x}}_l$ for each $l$ with feature-wise non-linearities. We learn a non-linear **SPAM** $P(\cdot; \boldsymbol{\theta})$ as,

$$P(\boldsymbol{x}; \boldsymbol{\theta}) = b + \langle \boldsymbol{u}_1, F_1(\boldsymbol{x}) \rangle + \sum_{i=1}^{r_2} \lambda_{2i} \cdot \langle \boldsymbol{u}_{2i}, F_2(\boldsymbol{x}) \rangle^2 + ... + \sum_{i=1}^{r_k} \lambda_{ki} \cdot \langle \boldsymbol{u}_{ki}, F_k(\boldsymbol{x}) \rangle^k.$$

Here, the function $F_i(\boldsymbol{x}) = [f_{i1}(x_1), f_{i2}(x_2), ..., f_{id}(x_d)]$ is a Neural Additive Model (NAM) [Agarwal et al., 2021]. We denote this model as SPAM-NEURAL. For more details on the NAM we use, please refer to the Appendix Section C. Note that the resulting model is still interpretable, as the interaction between features $x_i$ and $x_j$, e.g., will be given by $(\sum_{k=1}^{r_2} \lambda_{2k} u_{2ki} u_{2kj}) \cdot f_{2i}(x_i) f_{2j}(x_j)$ instead of the previous $(\sum_{k=1}^{r_2} \lambda_{2k} u_{2ki} u_{2kj}) \cdot \sqrt{x_i \cdot x_j}$ for typical SPAM.

**Dropout for bases via $\lambda$.** To capture non-overlapping feature correlations, we introduce a *basis dropout* by setting the contribution from any particular basis to zero at random via the singular values $\lambda$. Specifically, observe that the contribution of any basis-singular value pair $(\boldsymbol{u}, \lambda)$ at order $l$ is $\lambda \langle \boldsymbol{u}, \boldsymbol{x} \rangle^l$. We apply dropout to $\lambda$ to ensure that the network learns robust basis directions.

### 3.3 Approximation and Learning-Theoretic Guarantees

We present some learning-theoretic guarantees for SPAM. Proofs are deferred to Appendix Section A.

**Proposition 2.** *Let $f$ be a continuous real-valued function over a compact set $\mathcal{X}$. For any threshold $\epsilon > 0$, there exists a low-rank polynomial $P : \mathcal{X} \to \mathbb{R}$ such that $\sup_{\boldsymbol{x} \in \mathcal{X}} |f(\boldsymbol{x}) - P(\boldsymbol{x})| < \epsilon$.*

The above result demonstrates that asymptotically, decomposed polynomials are universal function approximators. Next, we present an assumption on the data-generating distribution as well as a novel excess risk bound that characterizes learning with polynomials more precisely (non-asymptotic).

**Assumption 1** (Exponential Spectral Decay of Polynomial Approximators). *Let $\mathcal{P}_k$ denote the family of all polynomials of degree at most $k$, and let $P_{\star,k}$ denote the optimal polynomial in $\mathcal{P}_k$, i.e., $P_{\star,k} = \arg\min_{P \in \mathcal{P}_k} \mathbb{E}_{(\boldsymbol{x},y) \sim \mathfrak{P}} [\ell(P(\boldsymbol{x}), y)]$. We assume that $\forall k$, $P_{\star,k}$ admits a decomposition as described in Equation 1 such that, for all $1 \le l \le k$, there exist absolute constants $C_1 < 1$ and $C_2 = \mathcal{O}(1)$ such that $|\lambda_{lj}| \le C_1 \cdot \exp(-C_2 \cdot j^\gamma)$ for each $j \ge 1$ and $l \in [1, k]$.*

This condition is essentially bounding the "smoothness" of the nearest polynomial approximator of $f$, i.e., implying that only a few degrees of freedom suffice to approximate $f$ accurately. Assumption 1 provides a soft threshold for singular value decay. One can also replace the exponential with a slower "polynomial" decay with similar results, and we discuss this in the Appendix Section A. We can now present our error bound for $L_2$ regularized models (see Appendix for $L_1$-regularized models).

**Theorem 1.** *Let $\ell$ be 1-Lipschitz, $\delta \in (0,1]$, the generalization error for the optimal degree-$k$ polynomial $P_{\star,k}$ be $\mathcal{E}(P_{\star,k})$ and the generalization error for the empirical risk minimizer SPAM $\widehat{P}_{\boldsymbol{r},k}$, i.e., $\widehat{P}_{\boldsymbol{r},k} = \arg\min_P \sum_{i=1}^n \ell(P(\boldsymbol{x}_i), y_i)$ with ranks $\boldsymbol{r} = [1, r_2, ..., r_k]$ be $\mathcal{E}(\widehat{P}_{\boldsymbol{r},k})$. Then, if $\mathfrak{P}$ satisfies Assumption 1 with constants $C_1, C_2$ and $\gamma$, we have for $L_2-$regularized ERM, where $\|\boldsymbol{u}_{ij}\|_2 \le B_{u,2} \forall i \in [k], j \in [r_i]$, and $\|\boldsymbol{\lambda}\|_2 \le B_{\lambda,2}$ where $\boldsymbol{\lambda} = \{\{\lambda_{ij}\}_{j=1}^{r_i}\}_{i=1}^k$, with probability at least $1 - \delta$,*

$$\mathcal{E}(\widehat{P}_{\boldsymbol{r},k}) - \mathcal{E}(P_{\star,k}) \le 2B_{\lambda,2} \left( \sum_{l=1}^k (B_{u,2})^l \sqrt{r_l} \right) \sqrt{\frac{d}{n}} + \frac{C_1}{C_2} \cdot \left( \sum_{i=2}^k \exp(-r_i^\gamma) \right) + 5\sqrt{\frac{\log(4/\delta)}{n}}.$$

The above result demonstrates an expected scaling of the generalization error, matching the bounds obtained for $L_1$ (see Appendix Section A) and $L_2-$regularized linear classifiers [Wainwright, 2019]. We that the rank $r = \|\boldsymbol{r}\|_1$ has an $\mathcal{O}(r)$ dependence on the Rademacher complexity. This highlights a trade-off between optimization and generalization: a larger $r$ gives a lower approximation error (smaller second term), but a larger model complexity (larger first term). Observe, however, that even when $r = \Omega((\log d)^p)$ for some $p \ge 1$, the approximation error (second term) diminishes as $o(\frac{1}{d^{p\gamma}})$. This suggests that in practice, one only needs a cumulative rank (poly)logarithmic in $d$. The above result also provides a non-asymptotic variant of Proposition 2; as $n, r \to \infty$, $\mathcal{E}(\widehat{P}_{\boldsymbol{r},k}) \to \mathcal{E}(P_{\star,k})_+$.

## 4 Offline Experiments

Here we evaluate SPAM against a set of benchmark algorithms on both classification and regression problems. The baseline approaches we consider are (see Appendix Section C for precise details):

- **Deep Neural Networks (DNN)**: These are standard multi-layer fully-connected neural networks included to demonstrate an upper-bound on performance. These are explained via the perturbation-based LIME [Ribeiro et al., 2016] and SHAP [Lundberg and Lee, 2017] methods in Section 5.
- **Linear / Logistic Regression**: These serve as our baseline interpretable models. We learn both $L_1$ and $L_2-$regularized models on unary and pairwise features ($\binom{d}{2}$ features) with minibatch SGD.

Table 1: Evaluation of SPAM on benchmarks against prior work. (↑): higher is better, (↓): lower is better, **: variance across trials is $< 0.001$, * : variance across trials is $< 0.005$. **Boldface red** denotes the best **black-box** model, **boldface green** denotes the best **interpretable** model, **boldface** denotes where SPAM Order 3 is best. Runs averaged over 10 random trials with optimal hyperparameters.

| Model | Regression RMSE (↓) | Binary AUROC (↑) | Multi-Class Classification Accuracy (↑) | | | | Obj. Det. mAP (↑) |
|---|---|---|---|---|---|---|---|
| | CH | FICO | CovType | News | CUB | iNat | CO114 |
| *Interpretable Baselines* | | | | | | | |
| Linear (Order 1) | 0.7354** | 0.7909** | 0.7254** | 0.8238** | 0.7451** | 0.3932** | 0.1917** |
| Linear (Order 2) | 0.7293** | 0.7910** | 0.7601** | — | 0.7617** | 0.4292** | 0.2190** |
| EBMs (Order 1) | 0.5586** | 0.7985** | 0.7392** | — | — | — | — |
| EB$^2$Ms (Order 2) | 0.4919** | 0.7998** | — | — | — | — | — |
| NAM | 0.5721* | 0.7993** | 0.7359** | — | 0.7632** | 0.4194** | 0.2056** |
| *Uninterpretable Black-Box Baselines* | | | | | | | |
| XGBoost | **0.4428** | 0.7925** | 0.8860** | 0.7677** | 0.7186** | — | — |
| DNN | 0.5014** | **0.7936** | **0.9694** | **0.8494** | **0.7684** | **0.4584** | **0.2376** |
| *Our Interpretable Models* | | | | | | | |
| SPAM (Linear, Order 2) | 0.6474** | 0.7940** | 0.7732** | **0.8472** | **0.7786** | 0.4605** | **0.2361** |
| SPAM (Neural, Order 2) | **0.4914** | 0.8011* | **0.7770** | — | 0.7762** | **0.4689** | — |
| SPAM (Linear, Order 3) | 0.6410** | 0.7945** | 0.8066** | **0.8520** | 0.7741** | 0.4684** | **0.2368** |
| SPAM (Neural, Order 3) | 0.4865* | **0.8024** | 0.8857** | — | 0.7753** | **0.4722** | — |

Table 2: Tabular Datasets

| Name | California Housing (CH) | FICO | Cover Type (CovType) | Newsgroups |
|---|---|---|---|---|
| Source | Pace and Barry [1997] | FICO [2018] | Blackard and Dean [1999] | Lang [1995] |
| Instances | 20,640 | 10,459 | 581,012 | 18,828 |
| Features | 8 | 23 | 54 | 146,016 |
| Classes | - | 2 | 7 | 20 |
| Feature Type | Numeric | Mixed | Mixed | TF-IDF |

- **Gradient Boosted Trees (XGBoost)**: We use the library `xgboost`. This baseline is mainly to compare accuracy, as the number of trees required are typically large and are hence uninterpretable.
- **Explainable Boosting Machines (EBMs)** [Lou et al., 2013]: EBMs use millions of shallow bagged trees operating on each feature at a time. Note that this approach is not scalable to datasets with many features or multi-class problems. We report scores on the datasets where we successfully trained EBMs without sophisticated engineering, using the `interpretml` library [Nori et al., 2019].
- **Neural Additive Models (NAMs)** [Agarwal et al., 2021]: These models are neural network extensions of prior EBMs. Note that this method also does not scale to some datasets.

**Training Setup**. SPAM is implemented by learning $L_1/L_2$-regularized variants by minibatch SGD implemented in PyTorch. For regression tasks, we measure the root mean squared error (RMSE). For binary classification, we report the area under the ROC (AUROC), for multi-class classification, we report the top-1 accuracy (Acc@1), and finally, for object detection, we report mean Average Precision (mAP). We tune hyperparameters via random sampling approach over a grid. Note that for all experiments, both NAM and SPAM-NEURAL have identical MLP structures to ensure identical approximation power. For definitions of metrics and hyperparameter ranges, see Appendix Section C.

## 4.1 Measuring Benchmark Performance

We select tasks to explore a variety of settings from regression to multi-class classification, and also explore different dataset scales, from a few hundred samples and tens of features to 100K-scale datasets (both in the number of samples and data dimensionality), while ensuring that the features are interpretable. Our datasets are summarized in Table 2. Please see Appendix Section C.1 for details. For all datasets with no defined train-val-test split, we use a fixed random sample of 70% of the data for training, 10% for validation and 20% for testing. For the 20 Newsgroups dataset, we split the pre-defined training split 7:1 for training and validation, respectively.

In an effort to scale interpretable approaches beyond tabular datasets, we consider benchmark problems using the "Independent Concept Bottleneck" framework of Koh et al. [2020] for image

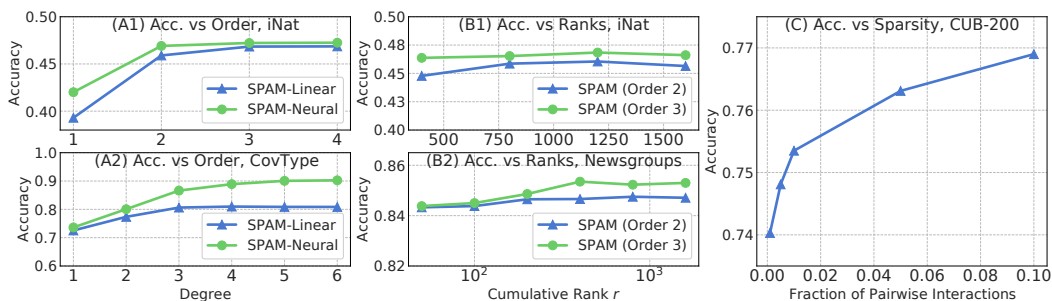

Figure 1: Ablation of Accuracy with: (A1, A2) Degree $k$; (B1, B2) Rank $r$; (C) $L_1$ Sparsity.

classification and object detection. We use a convolutional neural network (CNN) **backbone** ResNet-50 [He et al., 2016] that is trained to predict *interpretable concepts* (e.g., parts of object) from images. After training the **backbone**, we extract predicted concepts for all inputs, and learn an interpretable classifier **head** from these concepts. In experiments, the backbone remains identical for all comparisons, and we only compare the **head** classifiers (see Appendix Section C.3 for more details). We select three datasets for evaluation in the concept bottleneck setting:

1. **Caltech-UCSD Birds (CUB-200)** [Wah et al., 2011]: This is a fine-grained visual categorization dataset where one has to classify between 200 species of birds. The interpretable concepts are 278 *binary* bird attributes, e.g., the shape of the beak, the color of the wings, etc. The dataset has 5,994 training and 5,794 validation images. See Appendix Section C.3.1 for more details.
2. **iNaturalist "Birds"** [Van Horn et al., 2018, 2021]: iNaturalist can be thought of as a larger version of the previous dataset. We only select the "Birds" super-category of samples, which has 414,000 training and 14,860 validation instances across 1,486 classes. Since the iNaturalist dataset does not contain dense part and attribute annotations, we use the *predicted* concepts from the CUB-200 backbone model extracted on all samples.
3. **Common Objects Dataset** (CO114, Proprietary): Here we consider, for the first time, a concept bottleneck model for object detection. We construct a dataset involving common household objects (e.g., bed, saucer, tables, etc.) with their bounding box annotations. For each bounding box, we collect 2,618 interpretable annotations (e.g., parts, color, etc.). The dataset has 2,645,488 training and 58,525 validation samples across 115 classes with 2,618 interpretable concepts. We report the mean Average Precision (mAP) metric. For more details please refer to Appendix Section C.3.2.

Our results are summarized in Table 1. There are four main takeaways from the results. First, observe that both SPAM-LINEAR and SPAM-NEURAL comfortably outperform their prior interpretable counterparts on all datasets (e.g., Order 2 SPAM-LINEAR outperforms all linear methods – even the full rank pairwise model), and SPAM-NEURAL comfortably outperforms all non-linear baselines. Next, observe that in all datasets but CoverType, second degree interactions suffice to match or even outperform DNN performance. We will discuss CoverType in detail in the next paragraph. Thirdly, observe that SPAM models are as scalable as the black-box approaches, whereas prior work (e.g., NAM and EBMs) are not. In the case of NAM, we were unable to scale due to the sheer number of parameters required to model large datasets, e.g., Newsgroups ($\approx$900M parameters), and for EBMs, the training time and memory requirement increased dramatically for large datasets. Furthermore, $EB^2Ms$ do not even support multi-class problems. Finally, observe that for many problems, we do not require *non-linear* feature transformations once we model feature interactions: e.g., iNat and CUB.

**CoverType Dataset.** XGBoost and DNNs perform substantially better on CoverType compared to linear or neural SPAM. On analysis, we found that existing SPAM models underfit on CoverType, and hence we increased the total parameters for SPAM-NEURAL via *subnets*, identical to NAMs [Agarwal et al., 2021], where each feature is mapped to $s$ non-linear features (as opposed to 1 originally). With $s = 8$ subnets, the performance of SPAM-NEURAL (order 3) improves to $0.9405$. In comparison, NAMs with $8$ subnets provides a lower accuracy of $0.7551$.

## 4.2 Ablation Studies

**Rank and Degree of Interactions**. By Proposition 2, it is natural to expect the approximation quality to increase with the degree $k$ and rank $r$, i.e., as we introduce more higher-order interactions

Table 3: Throughput benchmarking (higher is faster).

| Method | CH | FICO | CovType | News |
|---|---|---|---|---|
| | Throughput (images / second) | | | |
| NAM [Agarwal et al., 2021] | $5 \times 10^5$ | $1.2 \times 10^5$ | $8 \times 10^4$ | 23 |
| NAM (Order 2) | $1.1 \times 10^4$ | $6 \times 10^3$ | $3 \times 10^3$ | - |
| LinearSPAM (Order 2) | $6.1 \times 10^7$ | $6.7 \times 10^7$ | $6.1 \times 10^7$ | $2.6 \times 10^6$ |
| NeuralSPAM (Order 2) | $1.7 \times 10^5$ | $7.9 \times 10^3$ | $4.1 \times 10^3$ | - |
| LinearSPAM (Order 3) | $3.2 \times 10^7$ | $3.7 \times 10^7$ | $3.9 \times 10^7$ | $1.8 \times 10^5$ |
| NeuralSPAM (Order 3) | $1.1 \times 10^5$ | $5.3 \times 10^3$ | $2.6 \times 10^3$ | - |
| MLP | $1.3 \times 10^7$ | $1.3 \times 10^7$ | $1.3 \times 10^7$ | $2.2 \times 10^5$ |

between features. We ablate the degree $k$ on the tabular benchmark **CoverType** and concept bottleneck benchmark **iNaturalist (Birds)**, as summarized in Figure 1A. We observe, as expected, that increasing the degree $k$ leads to moderate improvements beyond $k \geq 3$, but with diminishing returns. Similarly, we examine the effect of the cumulative rank $r$ on performance on the **Newsgroups** and **iNaturalist (Birds)** datasets (Figure 1B); we observe that performance is sufficiently insensitive to $r$, and plateaus after a while. We imagine that as $r$ increases, model complexity will dominate and performance will likely begin to decrease, matching the full-rank performance at $r = \mathcal{O}(d^2)$ for pairwise models.

**Sparsity in Higher-Order Relationships.** A requirement for interpretability is to ensure that the learned models can be explained with a few interpretable concepts. Since the number of higher-order combinations increases with $k$, we examine sparsity to limit the number of active feature assignments. We penalize the objective in Equation 2 with a regularization term that inhibits dense feature selection. If $\mathbf{U} = \{\{\boldsymbol{u}_{li}\}_{i=1}^{r_l}\}_{l=1}^{k}$ denotes all the basis vectors in matrix form, we add the penalty $\mathcal{R}(\boldsymbol{\theta}) \triangleq \|\mathbf{U}\|_1$. This ensures that every basis $\boldsymbol{u}$ only captures limited interactions between features, and hence the overall complexity of the model is limited. We examine accuracy as a function of the fraction of non-zero pairwise interactions for a degree 2, rank 800 SPAM on CUB-200 in Figure 1C, and find that only $6\%$ of the possible interactions suffice to obtain competitive performance.

**Examining Assumption 1 (Spectral Decay).** The spectral decay assumption presented in Assumption 1 is crucial to obtain reasonable generalization bounds (Theorem 1). To test this assumption, we examine the spectra of a SPAM-LINEAR (Order 2) model for all 200 of CUB-200 classes. These are depicted in Figure 2A. Furthermore, we fit an exponential model on the decay itself, and obtain (in line with Assumption 1) $\gamma = 3$, $C_1 = 0.54$ and $C_2 = 0.006$, all in accordance with the assumption.

## 4.3 Additional Experiments

### 4.3.1 Comparisons on Common Benchmarks

In addition to the 7 datasets we considered earlier, we evaluate on 9 further benchmark datasets commonly used in the interpretability literature. We evaluate on the MIMIC2, Credit and COMPAS datasets as presented in Agarwal et al. [2021], and the Click, Epsilon, Higgs, Microsoft, Yahoo and Year datasets from Chang et al. [2021]. The result of this is summarized in Table 4. We observe that SPAM models are competitive with prior work, and outperforming prior work on a number of tasks (the best interpretable model is in **bold**).

### 4.3.2 Runtime Evaluation

We provide a comparison of SPAM runtimes on 4 different datasets to establish their scalability. We consider the California Housing (CH, Pace and Barry [1997], 8 features), FICO HELOC (FICO, FICO [2018], 23 features), CoverType (CovType, Blackard and Dean [1999], 54 features) and 20 Newsgroups (News, Lang [1995], 146,016 features), and evaluate the throughput of the optimal Linear and Neural SPAM models against Neural Additive Models (NAM, Agarwal et al. [2021]) and MLPs. Table 3 describes the throughput, where we observe that Linear SPAM is more efficient than MLPs, and NeuralSPAM is orders of magnitude faster than NAM, which does not scale to higher orders.

Table 4: Comparison against state-of-the-art interpretable neural networks on additional benchmarks.

| Method | MIMIC2 (AUC) | Credit (AUC) | COMPAS (AUC) | Click (ERR) | Epsilon (ERR) | Higgs (ERR) | Microsoft (MSE) | Yahoo (MSE) | Year (MSE) |
|---|---|---|---|---|---|---|---|---|---|
| NAM [Agarwal et al., 2021] | 0.8539 | 0.9766 | 0.7368 | 0.3447 | 0.1079 | 0.2972 | 0.5824 | 0.6093 | 85.25 |
| NODE-GAM [Chang et al., 2021] | 0.8320 | 0.9810 | 0.7420 | **0.3342** | 0.1040 | 0.2970 | 0.5821 | 0.6101 | 85.09 |
| LinearSPAM (Order 2) | 0.8514 | 0.9836 | **0.7426** | 0.3791 | **0.1011** | 0.2881 | 0.5710 | 0.5923 | 81.30 |
| NeuralSPAM (Order 2) | **0.8664** | **0.9850** | 0.7411 | 0.3348 | 0.1020 | **0.2750** | **0.5671** | **0.5869** | **79.99** |
| XGBoost | 0.8430 | 0.9780 | 0.7440 | 0.3334 | 0.1112 | 0.2328 | 0.5544 | 0.5420 | 78.53 |

Table 5: Feature importances for interpretable models used in human subject evaulations.

| Model | Importances for input $\boldsymbol{x} = \{x_1, ..., x_d\}$ |
|---|---|
| Linear / *post-hoc* Linear | $w_i \cdot x_i, i \in \{1, ..., d\}$ |
| SPAM (Linear, Order 2) | $u_{1i} \cdot x_i, i \in \{1, ..., d\}$ and $\left(\sum_{l=1}^{r_2} \lambda_{2l} u_{2li} u_{2lj}\right) \cdot \sqrt{x_i \cdot x_j}, i, j \in [d]$ |

## 5   Human Subject Evaluations

**Experiment Methodology**.    We now evaluate how well explanations from **SPAM** fare *in a practical setting* with non-interpretable benchmarks such as black-box models equipped with *post-hoc* explanations. Our objective is to mimic a practical setting, where the model user must decide between a fully-interpretable model vs. a black-box classifier with *post-hoc* explainability. Our experiment design is a Prediction Task [Hoffman et al., 2018, Muramatsu and Pratt, 2001], where the objective for the users is to guess what the model predicted given an explanation. The motivation of such a design is to ascertain both the *faithfulness* and *interpretability* of the model explanations.

**Computing Explanations**. We first compute the feature importances for any class $c$, as the *contribution* of any specific feature in the logit for class $c$ (for binary classification or regression, there is only one class). For the Logisitc Regression model, we simply use the contribution of each feature to the prediction. For SPAM-Linear (Order 2), we still use each feature's contribution, but features now can be unary ($x_i$) or pairwise ($x_i$ and $x_j$). As black-box *post-hoc* baselines, we consider DNNs with LIME [Ribeiro et al., 2016] and KernelSHAP [Lundberg and Lee, 2017] to generate *post-hoc* linear explanations. See Table 5 for the formulation of importance for the models we use in experiments.

**Experiment Design** We conduct a separate experiment for each model **M** with explanations of length $E$ by selecting $E$ most important features. Each such experiment runs in two phases - *training* and *testing*. In the *training* phase, participants are shown 8 images of birds and their corresponding explanations to develop their mental model. Four of these images have been predicted as class **A** by the model, and the remaining are predicted as class **B**. They then move on to the *testing* phase, where they are successively shown the explanations for 5 unseen images (which the model could have predicted as either **A** or **B**), and the users must answer *"given the explanation, which class (of **A** or **B**) did the model predict?"*. If they desire, the users can move back to the training phase at any time to revise model explanations. We do not show the corresponding images in the test phase, as we want the user feedback to rely solely on the faithfulness of explanations. We measure the *mean user accuracy* on predicting model decisions. For more details on the interface, please see Appendix Section D.1. For each experiment, we follow an independent-subjects design, and gathered a total of 150 participants per experiment. Each task (corresponding to one model-explanation pair) lasted 3 minutes. The participants were compensated with $0.75 USD ($15/hour), and all experiments were run on Amazon Mechanical Turk (AMT, Buhrmester et al. [2016]) with a total cost of $3000. To remove poorly performing participants, we only select those that get the first decision correct, and compute the mean user accuracy using the remaining 4 images.

**Study #1: Comparing Black-Box and Transparent Explanations**. We compare different interpretable models with a *fixed* explanation length of $E = 7$. Our objective is to assess the interpretability of pairwise interactions compared to black-box and linear explanations, and to re-answer whether it was even necessary to have fully-interpretable models compared to black-box models with *post-hoc* explanations. We used samples from the 5 pairs of classes, that is **A** and **B**), from CUB-200 dataset (see Section D.1 for images). The results from this experiment are summarized in Figure 2B. Observe that with SPAM, the mean user accuracy (mUA) is substantially higher (0.71) compared to both linear (0.67) and *post-hoc* LIME (0.65). We would like to remark that even for $E = 7$, SPAM has an

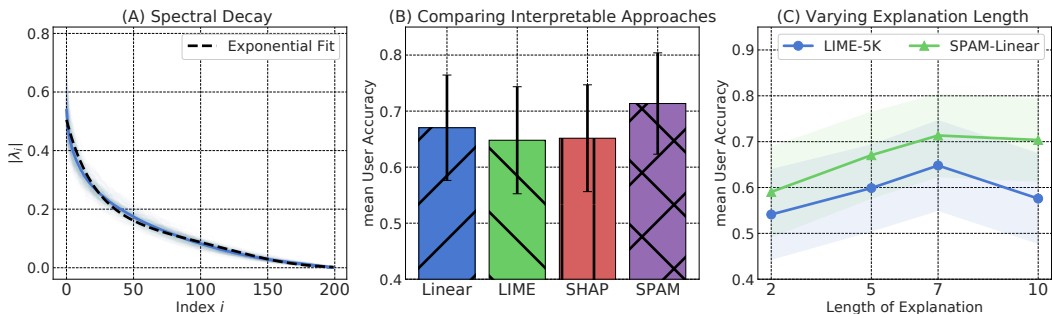

Figure 2: (A) Spectral Decay on CUB-200; Human Subject Evaluation results for (B) Comparing Black-box and glass-box explanations, and (C) LIME vs. SPAM with different explanation lengths.

average of 3.1 pairwise terms in each explanation, and therefore the higher-order terms are substantial. A one-sided t-test [Freund and Simon, 1967] provides a significance ($p-$value) of $2.29 \times 10^{-4}$ and a test statistic of 3.52 for the SPAM mUA being higher than LIME, and correspondingly for Linear, we get a significance ($p-$value) of 0.03801 and a test statistic of 1.666. Hence SPAM's improvements in interpretability are statistically significant over both linear and *post-hoc* approaches.

**Study #2: Varying Explanation Length**. One can argue that the increased interpretability of polynomial approaches is due to the fact that each higher-order "term" involves multiple features (e.g., pairwise involves two features), and hence the increased mean user accuracy (mUA) is due to the larger number of features shown (either individually or as pairs). We examine this hypothesis by varying the explanation length $E$ for both SPAM-Linear (Order 2) and LIME, which consequently increases the number of terms seen for both approaches. We observe in Figure 2C that regardless of explanation method, mUA is maximum at $E = 7$, and introducing more terms decreases interpretability.

## 6 Discussion and Conclusion

We presented a simple and scalable approach for modeling higher-order interactions in interpretable machine learning. Our approach alleviates several of the concerns with existing algorithms and focuses on presenting a viable alternative for practical, large-scale machine learning systems. In addition to offline experimental guarantees, we demonstrate by an extensive user study that these models are indeed more interpretable in practice, and therefore can readily substitute *post-hoc* interpretability without any loss in performance. Moreover, to the best of our knowledge, our work provides the first incremental analysis of interpretability and performance: we show how progressively increasing the model complexity (with higher order interactions) brings better performance while compromising interpretability in practice, and the "sweet spot" appears to rest at pairwise interactions.

We have several follow-up directions. First, given that our model provides precise guarantees on generalization, it is a feasible starting point to understanding tradeoffs between privacy and explainability from a rigorous perspective. Next, one can consider utilizing SPAM to understand failure modes and spurious correlations directly. Furthermore, SPAM-esque decompositions can also be useful in other domains beyond interpretability, e.g., language and vision, where modeling higher-order interactions is challenging due to the curse of dimensionality.

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
