# A Deferred Proofs

## A.1 Proof of Proposition 1

This follows from noting that the Hessian of the objective function is positive definite for all $\lambda > 0$ when the constraints in the proposition hold.

## A.2 Proof of Proposition 2

This follows directly from the Stone-Weierstrass Theorem [Stone, 1948, Weierstrass, 1885] and Theorem 1 of Chui and Li [1993].

## A.3 Proof of Theorem 1

We state Theorem 1 in full here.

**Theorem 2.** *Let $\ell$ be a 1-Lipschitz loss, $\delta \in (0,1]$ and the generalization error for the optimal degree $k$ polynomial $P_{\star,k}$ be given by $\mathcal{E}(P_{\star,k})$ and the training error for the ERM SPAM $\widehat{P}_{\boldsymbol{r},k}$, i.e.,* $\widehat{P}_{\boldsymbol{r},k} = \arg\min_P \sum_{i=1}^n \ell(P(\boldsymbol{x}_i, y_i)$ *with ranks $\boldsymbol{r} = [1, r_2, ..., r_k]$ be given by $\widehat{\mathcal{E}}_n(\widehat{P}_{\boldsymbol{r},k})$. Then, for $L_2-$regularized models, where $\|\boldsymbol{u}_{ij}\|_2 \le B_{u,2}$ for all $i \in [k], j \in [r_i]$, and $\|\boldsymbol{\lambda}\|_2 \le B_{\lambda,2}$ where $\boldsymbol{\lambda} = \{\{\lambda_{ij}\}_{j=1}^{r_i}\}_{i=1}^k$, we have that with probability at least $1 - \delta$ there exists an absolute constant $C$ such that,*

$$\mathcal{E}(\widehat{P}_{\boldsymbol{r},k}) - \mathcal{E}(P_{\star,k}) \le 2B_{\lambda,2}\left(\sum_{l=1}^k (B_{u,2})^l \sqrt{r_l}\right)\sqrt{\frac{d}{n}} + C \cdot \left(\sum_{i=2}^k \exp(-r_i^\gamma)\right) + 5\sqrt{\frac{\log(4/\delta)}{n}}.$$

*For $L_1-$regularization, $\|\boldsymbol{u}_{ij}\|_1 \le B_{u,1}$ for all $i \in [k], j \in [r_i]$, and $\|\boldsymbol{\lambda}\|_1 \le B_{\lambda,1}$ where $\boldsymbol{\lambda} = \{\{\lambda_{ij}\}_{j=1}^{r_i}\}_{i=1}^k$, we have with probability at least $1 - \delta$ there exists an absolute constant $C$ such that,*

$$\mathcal{E}(\widehat{P}_{\boldsymbol{r},k}) - \mathcal{E}(P_{\star,k}) \le 2B_{\lambda,1}\left(\sum_{l=1}^k (B_{u,1})^l\right)\sqrt{\frac{\log(d)}{n}} + C \cdot \left(\sum_{i=2}^k \exp(-r_i^\gamma)\right) + 5\sqrt{\frac{\log(4/\delta)}{n}}.$$

While we will eventually prove Theorem 1, the approach is to first establish the necessary mathematical background and outline the proof approach. We will provide a general result under arbitrary Lipschitz loss functions via a metric entropy bound for low-rank polynomial approximations. We will denote, for any $k > 1$, the space of all *low-rank* polynomials of order $k$ with cumulative rank $\boldsymbol{r} = [1, r_1, r_2, ..., r_k] \in \mathbb{R}^k$ as $\boldsymbol{\Theta}_{\boldsymbol{r}} \subset \mathbb{R}_+^{d(r+1)}$ where $r = \|\boldsymbol{r}\|_1$. Recall from Section 3 that any $\boldsymbol{\theta} \in \boldsymbol{\Theta}$ is composed of the components, $\{b, \{\{\lambda_{ij}\}_{j=1}^{r_i}\}_{i=1}^k, \{\{\boldsymbol{u}_{ij}\}_{j=1}^{r_i}\}_{i=1}^k\}$ and we use $\boldsymbol{\theta}$ as a shorthand.

Now, for any function $f : \mathcal{X} \to \mathcal{Y}$ and any bounded $L-$Lipschitz loss function $\ell : \mathcal{Y} \times \mathcal{Y} \to [0,1]$ (we assume, without loss of generality, that the range of $\ell$ is the closed interval $[0,1]$), the *training error* over $n$ samples as,

$$\widehat{\mathcal{E}}_n(f) = \frac{1}{n}\sum_{i=1}^n [\ell(f(\boldsymbol{x}_i), y_i)].$$

Similarly, we can define the *expected risk* over the sample distribution $\mathfrak{P} : \mathcal{X} \times \mathcal{Y} \to [0, 1)$ as,

$$\mathcal{E}(f) = \mathbb{E}_{(\boldsymbol{x},y)\sim\mathfrak{P}}[\ell(f(\boldsymbol{x}), y)].$$

At a high level, our objective is to bound the *expected risk* of the ERM *low-rank decomposed polynomial* (LRDP) $\mathcal{E}(\widehat{P}_{\boldsymbol{r},k})$ of some degree $k$ and (cumulative) rank $r$ with risk of the *optimal* polynomial of degree $k$, i.e., $\mathcal{E}(P_{\star,k})$. Note that $P_{\star,k}$ is not necessarily low-rank and can have cumulative rank of order $\mathcal{O}(d^k)$. Let us denote the maximum possible rank of any polynomial of degree $k$ as $\bar{r}$ and the corresponding set of ranks as $\bar{\boldsymbol{r}}$. Observe that any polynomial of degree $k$ has a (non-unique) representation in $\boldsymbol{\Theta}_{\bar{\boldsymbol{r}}}$, and furthermore,

$$P_{\star,k} = \arg\min_{\boldsymbol{\theta}\in\boldsymbol{\Theta}_{\bar{\boldsymbol{r}}}}(\mathcal{E}(P(\cdot;\boldsymbol{\theta}))). \tag{4}$$

Similarly, we have that the *empirical risk minimizer* LRDP $\widehat{P}_{\boldsymbol{r},k}$ of degree $k$ and ranks $\boldsymbol{r}$ satisfies,

$$\widehat{P}_{\boldsymbol{r},k} = \arg\min_{\boldsymbol{\theta}\in\boldsymbol{\Theta}_{\boldsymbol{r}}}\left(\widehat{\mathcal{E}}_n(P(\cdot;\boldsymbol{\theta}))\right). \tag{5}$$

We can now begin the proof. For the optimal polynomial $P_{\star,k}$, let us denote the corresponding singular values as $\{\{\lambda_{ij}^\star\}_{j=1}^{\bar{r}_i}\}_{i=1}^k$, bases as $\{\{\boldsymbol{u}_{ij}^\star\}_{j=1}^{\bar{r}_i}\}_{i=1}^k$ and bias as $b_{\star,k}$. We will bound the *excess risk* $\mathcal{E}(\widehat{P}_{\boldsymbol{r},k}) - \mathcal{E}(P_{\star,k})$ by introducing a third polynomial $\widetilde{P}_{\boldsymbol{r},k} \in \boldsymbol{\Theta}_{\boldsymbol{r}}$ which is defined as a *truncated* version of $P_{\star,k}$ up to the ranks $\boldsymbol{r}$. Specifically, we set the singular values $\{\{\widetilde{\lambda}_{ij}\}_{j=1}^{r_i}\}_{i=1}^k$,

bases as $\{\{\widetilde{\boldsymbol{u}}_{ij}\}_{j=1}^{r_i}\}_{i=1}^k$ and bias as $\widetilde{b}_k$ as,

$$\text{For } 1 \le j \le r_i, 1 \le i \le k, \ \ \widetilde{\lambda}_{ij} = \lambda_{ij}^\star, \widetilde{\boldsymbol{u}}_{ij} = \boldsymbol{u}_{ij}^\star, \text{ and } \widetilde{b}_k = b_k^\star. \tag{6}$$

The polynomial $\widetilde{P}_{\boldsymbol{r},k}$ can therefore be thought of as a "truncated" version of the optimal polynomial $P_{\star,k}$ up to the ranks $\boldsymbol{r}$. Observe that we can then write the excess risk as,

$$\mathcal{E}(\widehat{P}_{\boldsymbol{r},k}) - \mathcal{E}(P_{\star,k}) = \underbrace{\mathcal{E}(\widehat{P}_{\boldsymbol{r},k}) - \widehat{\mathcal{E}}_n(\widehat{P}_{\boldsymbol{r},k})}_{\textcircled{A}} + \underbrace{\widehat{\mathcal{E}}_n(\widehat{P}_{\boldsymbol{r},k}) - \widehat{\mathcal{E}}_n(\widetilde{P}_{\boldsymbol{r},k})}_{\le 0} + \underbrace{\widehat{\mathcal{E}}_n(\widetilde{P}_{\boldsymbol{r},k}) - \mathcal{E}(P_{\star,k})}_{\textcircled{B}}.$$

The term $\widehat{\mathcal{E}}_n(\widehat{P}_{\boldsymbol{r},k}) - \widehat{\mathcal{E}}_n(\widetilde{P}_{\boldsymbol{r},k}) \le 0$ since $\widehat{P}_{\boldsymbol{r},k}$ minimizes the empirical risk within $\boldsymbol{\Theta}_{\boldsymbol{r}}$ (Equation 5). Hence bounding terms $\textcircled{A}$ and $\textcircled{B}$ will provide us the bound. We will first bound term $\textcircled{B}$ via Lemma 1. We have that with probability at least $1 - \delta/2$ for any $\delta \in (0, 1]$,

$$\left| \widehat{\mathcal{E}}_n(\widetilde{P}_{\boldsymbol{r},k}) - \mathcal{E}(P_{\star,k}) \right| \le LC \cdot \left( \sum_{i=2}^k \exp(-r_i^\gamma) \right) + 2\sqrt{\frac{\log(2/\delta)}{n}}.$$

To bound term $\textcircled{A}$, we proceed via bounding the Rademacher complexity, a classic approach in statistical learning [Wainwright, 2019]. We will first establish a bound on the Rademacher complexity of $\boldsymbol{\Theta}_{\boldsymbol{r}}$ under $L^1$ and $L^2-$regularization. Note that similar bounds via directly bounding the metric entropy or Dudley's entropy can also be obtained, but we present this approach for simplicity. We anticipate that unless alternative assumptions are made about the polynomials, the bounds will remain identical via these approaches. Observe first, that since the loss function is Lipschitz and bounded, we have with probability at least $1 - \delta/2, \delta \in (0, 1]$, from Theorem 4 and Theorem 8 of Bartlett and Mendelson [2002],

$$\mathcal{E}(\widehat{P}_{\boldsymbol{r},k}) - \widehat{\mathcal{E}}_n(\widehat{P}_{\boldsymbol{r},k}) \le \mathfrak{R}_n(\ell \odot \mathcal{P}(\boldsymbol{\Theta}_{\boldsymbol{r}})) + \sqrt{\frac{8\log(4/\delta)}{n}}. \tag{7}$$

Where $\mathfrak{R}_n$ denotes the empirical Rademacher complexity at $n$ samples [Bartlett and Mendelson, 2002], and $\mathcal{P}(\boldsymbol{\Theta}_{\boldsymbol{r}})$ denotes the set of all polynomials represented by the parameterization $\boldsymbol{\Theta}_{\boldsymbol{r}}$. Now, observe that any element in $\mathcal{P}(\boldsymbol{\Theta}_{\boldsymbol{r}})$ comprises $k + 1$ polynomials of a fixed degree (one for each degree from 0 to $k$). With some abuse of notation, let us denote the family of all polynomials with a degree exactly $l$ and rank exactly $r$ as $\mathcal{P}_{l,r}$. Observe that by Theorem 4,

$$\mathfrak{R}_n(\mathcal{P}(\boldsymbol{\Theta}_{\boldsymbol{r}})) \le \sum_{l=0}^k \mathfrak{R}_n(\mathcal{P}_{l,r_l}) = \sum_{l=1}^k \mathfrak{R}_n(\mathcal{P}_{l,r_l}). \tag{8}$$

The last equality follows from the fact that $\mathcal{P}_{0,\cdot}$ is always a constant function with a Rademacher complexity of 0. Furthermore, since $\ell$ is Lipschitz, we have once again by Theorem 4 that $\mathfrak{R}_n(\ell \odot \mathcal{P}(\boldsymbol{\Theta}_{\boldsymbol{r}})) \le 2L \cdot \mathfrak{R}_n(\mathcal{P}(\boldsymbol{\Theta}_{\boldsymbol{r}}))$. Putting this together, we have that with probability at least $1 - \delta/2, \delta \in (0, 1]$,

$$\mathcal{E}(\widehat{P}_{\boldsymbol{r},k}) - \widehat{\mathcal{E}}_n(\widehat{P}_{\boldsymbol{r},k}) \le 2L \cdot \sum_{l=1}^k \mathfrak{R}_n(\mathcal{P}_{l,r_l}) + \sqrt{\frac{8\log(4/\delta)}{n}}. \tag{9}$$

Now, to bound the remaining term, we consider two regularization constraints for $\boldsymbol{\Theta}_{\boldsymbol{r}}$. The first is $L_2-$regularization, where we assume that the parameter $\lambda, \boldsymbol{u}$ are bounded in $L_2$. Let $\boldsymbol{\lambda} = \{\{\lambda_{ij}\}_{j=1}^{r_i}\}_{i=1}^k$ denote the singular values across all ranks. Then, we assume for all $i \in [1, ..., k]$ and $j \in [1, ..., r_i]$ that $\|\boldsymbol{u}_{ij}\|_2 \le B_{u,2}$ and $\|\boldsymbol{\lambda}\|_2 \le B_{\lambda,2}$. Under this constraint, we can bound the empirical Rademacher complexity of each polynomial via Lemma 2. Replacing this result, we obtain with probability at least $1 - \delta/2, \delta \in (0, 1]$,

$$\mathcal{E}(\widehat{P}_{\boldsymbol{r},k}) - \widehat{\mathcal{E}}_n(\widehat{P}_{\boldsymbol{r},k}) \le 2LB_{\lambda,2}B_2^x \cdot \sum_{l=1}^k (B_{u,2})^l \sqrt{\frac{r}{n}} + \sqrt{\frac{8\log(4/\delta)}{n}}. \tag{10}$$

Where $B_2^x = \sup_{\boldsymbol{x} \in \mathcal{X}} \|\boldsymbol{x}\|_2$. Without loss of generality we can assume $B_2^x = \sqrt{d}$. Now, for $L_1-$regularized models, we assume that for all $i \in [1, ..., k]$ and $j \in [1, ..., r_i]$ that $\|\boldsymbol{u}_{ij}\|_1 \le B_{u,1}$ and $\|\boldsymbol{\lambda}\|_1 \le B_{\lambda,1}$. This gives us the following bound via Lemma 3:

$$\mathcal{E}(\widehat{P}_{\boldsymbol{r},k}) - \widehat{\mathcal{E}}_n(\widehat{P}_{\boldsymbol{r},k}) \le 2LB_{\lambda,1}B_\infty^x \cdot \sum_{l=1}^k (B_{u,1})^l \cdot \sqrt{\frac{\log(d)}{n}} + \sqrt{\frac{8\log(4/\delta)}{n}}. \tag{11}$$

Where $B_\infty^x = \sup_{\boldsymbol{x} \in \mathcal{X}} \|\boldsymbol{x}\|_\infty$. Without loss of generality we can assume $B_\infty^x = 1$. Using the bound for term $\textcircled{B}$ and applying a union bound provides us the final result.

### A.4 Polynomial Decay of Spectrum

As discussed earlier, one can replace the "exponential decay" assumption in Assumption 1 to a "softer" decay criterion with fatter tails, e.g., like a *polynomial* decay. We present extensions of earlier results to this polynomial decay condition. First we state the polynomial decay criterion precisely.

**Assumption 2.** *Let $\mathcal{P}_k$ denote the family of all polynomials of degree at most $k$, and let $P_{\star,k}$ denote the optimal polynomial in $\mathcal{P}_k$, i.e., $P_{\star,k} = \arg\min_{P \in \mathcal{P}_k} \mathbb{E}_{(\boldsymbol{x},y) \sim \mathfrak{P}}[\ell(P(\boldsymbol{x}), y)]$. We assume that $\forall k$, $P_{\star,k}$ admits a decomposition as described in Equation 1 such that, for all $1 \leq l \leq k$, there exist constants $C_1 < 1$ and $C_2 = \mathcal{O}(1)$ such that $|\lambda_{lj}| \leq C_1 \cdot j^{-\gamma-1}$ for each $j \geq 1$ and $l \in [1, k]$.*

**Theorem 3.** *Let $\ell$ be a 1-Lipschitz loss, $\delta \in (0, 1]$ and the generalization error for the optimal degree $k$ polynomial $P_{\star,k}$ be given by $\mathcal{E}(P_{\star,k})$ and the training error for the ERM SPAM $\widehat{P}_{\boldsymbol{r},k}$, i.e., $\widehat{P}_{\boldsymbol{r},k} = \arg\min_P \sum_{i=1}^n \ell(P(\boldsymbol{x}_i, y_i)$ with ranks $\boldsymbol{r} = [1, r_2, ..., r_k]$ be given by $\widehat{\mathcal{E}}_n(\widehat{P}_{\boldsymbol{r},k})$. Then, for $L_2-$regularized models, where $\|\boldsymbol{u}_{ij}\|_2 \leq B_{u,2}$ for all $i \in [k], j \in [r_i]$, and $\|\boldsymbol{\lambda}\|_2 \leq B_{\lambda,2}$ where $\boldsymbol{\lambda} = \{\{\lambda_{ij}\}_{j=1}^{r_i}\}_{i=1}^k$, we have that with probability at least $1 - \delta$ there exists an absolute constant $C$ such that,*

$$\mathcal{E}(\widehat{P}_{\boldsymbol{r},k}) - \mathcal{E}(P_{\star,k}) \leq 2B_{\lambda,2}\left(\sum_{l=1}^k (B_{u,2})^l \sqrt{r_l}\right)\sqrt{\frac{d}{n}} + C \cdot \left(\sum_{i=2}^k r_i^{-\gamma}\right) + 5\sqrt{\frac{\log(4/\delta)}{n}}.$$

*For $L_1-$regularization, $\|\boldsymbol{u}_{ij}\|_1 \leq B_{u,1}$ for all $i \in [k], j \in [r_i]$, and $\|\boldsymbol{\lambda}\|_1 \leq B_{\lambda,1}$ where $\boldsymbol{\lambda} = \{\{\lambda_{ij}\}_{j=1}^{r_i}\}_{i=1}^k$, we have with probability at least $1 - \delta$ there exists an absolute constant $C$ such that,*

$$\mathcal{E}(\widehat{P}_{\boldsymbol{r},k}) - \mathcal{E}(P_{\star,k}) \leq 2B_{\lambda,1}\left(\sum_{l=1}^k (B_{u,1})^l\right)\sqrt{\frac{\log(d)}{n}} + C \cdot \left(\sum_{i=2}^k r_i^{-\gamma}\right) + 5\sqrt{\frac{\log(4/\delta)}{n}}.$$

*Proof.* The proof is identical to the proof of Theorem 1, except for the application of Lemma 1. Note that following Lemma 1, the sum of the singular values can be bound using Assumption 2 as,

$$\sum_{i=1}^k \sum_{j=r_i}^{\bar{r}_i} |\lambda_{ij}^\star| \leq \sum_{i=1}^k \sum_{j=r_i}^{\bar{r}_i} C_1 \cdot j^{-\gamma-1} \leq \sum_{i=1}^k \int_{j=r_i}^\infty C_1 \cdot j^{-\gamma-1} d\gamma = \frac{C_1}{\gamma+1}\sum_{i=1}^k r_i^{-\gamma}.$$

Replacing this result completes the proof. □

**Discussion**. Observe that in contrast to the result for exponential decay (where one required $r$ to be polylogarithmic in $d$), we now require a larger rank $r$ to achieve competitive performance. Specifically, one can observe from Theorem 3 that setting $r_i = r/k = d^{\gamma/m}$ for some $m > 1$ guarantees that the error term decays at the rate $\mathcal{O}\left(k^{\frac{1-\gamma}{m}} d^{-\frac{1}{m}}\right)$ which diminishes quickly for small $m$ and large $d$. However, this degrades the rate of growth of the Rademacher complexity (first term) from a $\mathcal{O}(\sqrt{d})$ dependence to $\mathcal{O}(d^{\frac{1}{p}(1+\frac{\gamma}{m})})$ for $L_p$ regularization, $p \in \{1, 2\}$.

### A.5 Omitted Results

**Lemma 1.** *For the polynomials $\widetilde{P}_{\boldsymbol{r},k}$ and $P_{\star,k}$ as defined in Equations 4 and 6, the following holds with probability at least $1 - \delta, \delta \in (0, 1]$, for some absolute constant $C \ll 1$,*

$$\left|\widehat{\mathcal{E}}_n(\widetilde{P}_{\boldsymbol{r},k}) - \mathcal{E}(P_{\star,k})\right| \leq LC \cdot \left(\sum_{i=2}^k \exp(-r_i^\gamma)\right) + 2\sqrt{\frac{\log(2/\delta)}{n}}.$$

*Proof.* Observe,

$$\widehat{\mathcal{E}}_n(\widetilde{P}_{\boldsymbol{r},k}) - \mathcal{E}(P_{\star,k}) = \widehat{\mathcal{E}}_n(\widetilde{P}_{\boldsymbol{r},k}) - \mathcal{E}(\widetilde{P}_{\boldsymbol{r},k}) + \mathcal{E}(\widetilde{P}_{\boldsymbol{r},k}) - \mathcal{E}(P_{\star,k})$$

$$\leq \underbrace{\left|\widehat{\mathcal{E}}_n(\widetilde{P}_{\boldsymbol{r},k}) - \mathcal{E}(\widetilde{P}_{\boldsymbol{r},k})\right|}_{\text{B1}} + \underbrace{\left|\mathcal{E}(\widetilde{P}_{\boldsymbol{r},k}) - \mathcal{E}(P_{\star,k})\right|}_{\text{B2}}.$$

To bound B1, observe that for any point $(\boldsymbol{x}, y)$ within the training set, $\mathbb{E}[\ell(\widetilde{P}_{\boldsymbol{r},k}(\boldsymbol{x}), y)] = \mathcal{E}(\widetilde{P}_{\boldsymbol{r},k})$. Furthermore, $0 \leq \ell(\cdot, \cdot) \leq 1$. We can therefore apply the Azuma-Hoeffding inequality [Bercu et al., 2015] and obtain with probability at least $1 - \delta, \delta \in (0, 1]$,

$$\left|\widehat{\mathcal{E}}_n(\widetilde{P}_{\boldsymbol{r},k}) - \mathcal{E}(\widetilde{P}_{\boldsymbol{r},k})\right| \leq 2\sqrt{\frac{\log(2/\delta)}{n}}.$$

To bound $\boxed{B2}$, observe that since $\ell$ is $L-$Lipschitz, for some $x_1, x_2, y \in \mathcal{Y}$,
$$|\ell(x_1, y) - \ell(x_2, y)| \leq |L \cdot |x_1 - y| - L \cdot |x_2 - y||$$
$$\leq L \cdot |x_1 - x_2|.$$
Using this, we have that,
$$\left|\mathcal{E}(\widetilde{P}_{\boldsymbol{r},k}) - \mathcal{E}(P_{\star,k})\right| \leq \left|\mathbb{E}_{(\boldsymbol{x},y)\sim\mathfrak{P}}\left[\ell(\widetilde{P}_{\boldsymbol{r},k}(\boldsymbol{x}), y) - \ell(P_{\star,k}(\boldsymbol{x}), y)\right]\right|$$
$$\leq \mathbb{E}_{(\boldsymbol{x},y)\sim\mathfrak{P}}\left[\left|\ell(\widetilde{P}_{\boldsymbol{r},k}(\boldsymbol{x}), y) - \ell(P_{\star,k}(\boldsymbol{x}), y)\right|\right]$$
$$\leq L \cdot \mathbb{E}_{(\boldsymbol{x},y)\sim\mathfrak{P}}\left[\left|\widetilde{P}_{\boldsymbol{r},k}(\boldsymbol{x}) - P_{\star,k}(\boldsymbol{x})\right|\right]$$
$$\leq L \cdot \sup_{\boldsymbol{x}\in\mathcal{X}}\left|\widetilde{P}_{\boldsymbol{r},k}(\boldsymbol{x}) - P_{\star,k}(\boldsymbol{x})\right|.$$
Observe now that for any $\boldsymbol{x} \in \mathcal{X}$,
$$\left|P_{\star,k}(\boldsymbol{x}) - \widetilde{P}_{\boldsymbol{r},k}(\boldsymbol{x})\right| = \left|\sum_{i=1}^{k}\sum_{j=r_i}^{\bar{r}_i} \lambda_{ij}^\star \cdot \langle\boldsymbol{u}_{ij}^\star, \tilde{\boldsymbol{x}}_i\rangle^i\right| \leq \sum_{i=1}^{k}\sum_{j=r_i}^{\bar{r}_i} \left|\lambda_{ij}^\star \cdot \langle\boldsymbol{u}_{ij}^\star, \tilde{\boldsymbol{x}}_i\rangle^i\right| \leq \sum_{i=1}^{k}\sum_{j=r_i}^{\bar{r}_i}|\lambda_{ij}^\star|.$$
Recall that Assumption 1 can control $\sum_{i=1}^{k}\sum_{j=r_i}^{\bar{r}_i}|\lambda_{ij}^\star|$. If the spectrum obeys the $D-$finite condition and $r_i = D \forall i \in [1, k]$ we have that $\sum_{i=1}^{k}\sum_{j=r_i}^{\bar{r}_i}|\lambda_{ij}^\star| = 0$. Next, if the spectrum obeys the $\gamma-$exponential decay, we have that $\lambda_{ij} = C_1 \exp(-C_2 j^\gamma)$. This gives us,
$$\sum_{i=1}^{k}\sum_{j=r_i}^{\bar{r}_i}|\lambda_{ij}^\star| \leq \sum_{i=1}^{k}\sum_{j=r_i}^{\bar{r}_i} C_1 \exp(-C_2 j^\gamma) \leq \sum_{i=1}^{k}\int_{j=r_i}^{\infty} C_1 \exp(-C_2 j^\gamma).$$
By Equation E.16 from Yang et al. [2020] we can bound the R.H.S. since $\gamma \geq 1$,
$$\int_{j=r_i}^{\infty} C_1 \exp(-C_2 j^\gamma) \leq \frac{C_1}{C_2} \exp\left(-r_i^\gamma\right).$$
Replacing this result above completes the proof. $\qquad\square$

**Lemma 2** (Empirical Rademacher Complexity under $L_2$ regularization). *Let $\mathcal{P}_{l,r}$ denote the set of polynomials that have rank exactly $r$ and degree exactly $l$ such that they admit a rank-decomposed representation of $\sum_{j=1}^{r} \lambda_j \cdot \boldsymbol{u}_j \otimes ... \otimes \boldsymbol{u}_j$, where $\boldsymbol{\lambda} = [\lambda_1, ..., \lambda_r]$ s.t. $\|\boldsymbol{\lambda}\|_2 \leq B_{\lambda,2}$ and $\|\boldsymbol{u}_j\|_2 \leq B_{u,2}$ for all $j \in 1, ..., l$. Let the rescaled data distribution as defined in Equation 3 for degree $l$ be given by $\widetilde{\mathcal{X}}_l = \{sign(\boldsymbol{x}) \cdot |\boldsymbol{x}|^{1/l} | \boldsymbol{x} \in \mathcal{X}\}$ and we sample $n$ points i.i.d. from $\widetilde{\mathcal{X}}_l$. Then the empirical Rademacher complexity $\mathfrak{R}_n$ of $\mathcal{P}_{l,r}$ obeys,*
$$\mathfrak{R}_n(\mathcal{P}_{l,r}) \leq B_{\lambda,2} B_2^x (B_{u,2})^l \sqrt{\frac{r}{n}}.$$

*Proof.*
$$\mathfrak{R}_n(\mathcal{P}_{l,r}) = \frac{1}{n}\mathbb{E}\left[\sup_{\boldsymbol{\lambda}\in\boldsymbol{\Lambda},\mathcal{U}} \sum_{i=1}^{n} \epsilon_i \sum_{j=1}^{r} \lambda_r \cdot \langle\boldsymbol{u}_j, \tilde{\boldsymbol{x}}_{li}\rangle^l\right]$$
$$\leq \frac{1}{n}\sup_{\|\boldsymbol{\lambda}\|_2\leq B_{\lambda,2}} \sum_{j=1}^{r} \lambda_r \cdot \mathbb{E}\left[\sup_{\|\boldsymbol{u}_j\|_2\leq B_{u,2}} \sum_{i=1}^{n} \epsilon_i \cdot \langle\boldsymbol{u}_j, \tilde{\boldsymbol{x}}_{li}\rangle^l\right]$$
$$\leq \frac{1}{n}\sup_{\|\boldsymbol{\lambda}\|_2\leq B_{\lambda,2}} \|\boldsymbol{\lambda}\|_2 \cdot \sqrt{\sum_{j=1}^{r}\mathbb{E}\left[\sup_{\|\boldsymbol{u}_j\|_2\leq B_{u,2}} \sum_{i=1}^{n} \epsilon_i \cdot \langle\boldsymbol{u}_j, \tilde{\boldsymbol{x}}_{li}\rangle^l\right]^2}$$
$$\leq \frac{B_{\lambda,2}}{n}\sqrt{\sum_{j=1}^{r}\mathbb{E}\left[\sup_{\|\boldsymbol{u}_j\|_2\leq B_{u,2}} \sum_{i=1}^{n} \epsilon_n^2 \cdot \langle\boldsymbol{u}_j, \tilde{\boldsymbol{x}}_{li}\rangle^{2l}\right]}$$
$$\leq \frac{B_{\lambda,2}}{n}\sqrt{\sum_{j=1}^{r}\mathbb{E}\left[\sup_{\|\boldsymbol{u}_j\|_2\leq B_{u,2}} \sum_{i=1}^{n} \|\boldsymbol{u}_j\|_2^{2l}\|\tilde{\boldsymbol{x}}_{li}\|^{2l}\right]}$$
$$\leq B_{\lambda,2}(B_{u,2})^l B_2^x \sqrt{\frac{r}{n}}.$$

$\square$

**Lemma 3** (Empirical Rademacher Complexity under $L_1$ regularization)**.** *Let $\mathcal{P}_{l,r}$ denote the set of polynomials that have rank exactly $r$ and degree exactly $l$ such that they admit a rank-decomposed representation of $\sum_{j=1}^{r} \lambda_j \cdot \boldsymbol{u}_j \otimes ... \otimes \boldsymbol{u}_j$, where $\boldsymbol{\lambda} = [\lambda_1, ..., \lambda_r]$ s.t. $\|\boldsymbol{\lambda}\|_1 \le B_{\lambda,1}$ and $\|\boldsymbol{u}_j\|_1 \le B_{u,1}$ for all $j \in 1, ..., l$. Let the rescaled data distribution as defined in Equation 3 for degree $l$ be given by $\widetilde{\mathcal{X}}_l = \{sign(\boldsymbol{x}) \cdot |\boldsymbol{x}|^{1/l} | \boldsymbol{x} \in \mathcal{X}\}$ and we sample $n$ points i.i.d. from $\widetilde{\mathcal{X}}_l$. Then the empirical Rademacher complexity $\mathfrak{R}_n$ of $\mathcal{P}_{l,r}$ obeys,*

$$\mathfrak{R}_n(\mathcal{P}_{l,r}) \le (B_{\lambda,1} \cdot r)(B_{u,1})^l B_1^x \sqrt{\frac{\log(d)}{n}}.$$

*Proof.*

$$
\begin{aligned}
\mathfrak{R}_n(\mathcal{P}_{l,r}) &= \frac{1}{n} \mathbb{E}\left[ \sup_{\boldsymbol{\lambda}, \boldsymbol{u}} \sum_{i=1}^{n} \epsilon_i \sum_{j=1}^{r} \lambda_r \cdot \langle \boldsymbol{u}_j, \tilde{\boldsymbol{x}}_{li} \rangle^l \right] \\
&\le \frac{1}{n} \sup_{\|\boldsymbol{\lambda}\|_1 \le B_{\lambda,1}} \sum_{j=1}^{r} \lambda_r \cdot \mathbb{E}\left[ \sup_{\|\boldsymbol{u}_j\|_1 \le B_{u,1}} \sum_{i=1}^{n} \epsilon_i \cdot \langle \boldsymbol{u}_j, \tilde{\boldsymbol{x}}_{li} \rangle^l \right] \\
&\le \frac{1}{n} \sup_{\|\boldsymbol{\lambda}\|_1 \le B_{\lambda,1}} \|\boldsymbol{\lambda}\|_1 \cdot \max_{j \in [r]} \left| \mathbb{E}\left[ \sup_{\|\boldsymbol{u}_j\|_1 \le B_{u,1}} \sum_{i=1}^{n} \epsilon_i \cdot \langle \boldsymbol{u}_j, \tilde{\boldsymbol{x}}_{li} \rangle^l \right] \right| \\
&\le \frac{B_{\lambda,1}}{n} \max_{j \in [r]} \left[ \mathbb{E}\left[ \left| \sup_{\|\boldsymbol{u}_j\|_1 \le B_{u,1}} \sum_{i=1}^{n} \epsilon_i \cdot \langle \boldsymbol{u}_j, \tilde{\boldsymbol{x}}_{li} \rangle^l \right| \right] \right] \\
&\le \frac{B_{\lambda,1}}{n} \mathbb{E}\left[ \max_{j \in [r]} \left| \sup_{\|\boldsymbol{u}_j\|_1 \le B_{u,1}} \sum_{i=1}^{n} \epsilon_i \cdot \sum_{j=1}^{r} \langle \boldsymbol{u}_j, \tilde{\boldsymbol{x}}_{li} \rangle^l \right| \right] \\
&= \frac{B_{\lambda,1}}{n} \mathbb{E}\left[ \left| \sup_{\|\boldsymbol{u}_j\|_1 \le B_{u,1}} \sum_{i=1}^{n} \max_{j \in [r]} \|\boldsymbol{u}_j\|_1^l \|\epsilon_i \tilde{\boldsymbol{x}}_{li}\|_\infty^l \right| \right] \\
&\le \frac{(B_{u,1})^l \cdot B_{\lambda,1}}{n} \mathbb{E}\left[ \sum_{i=1}^{n} \|\epsilon_i \boldsymbol{x}_i\|_\infty \right] \\
&\le B_{\lambda,1}(B_{u,1})^l B_\infty^x \cdot \sqrt{\frac{\log(d)}{n}}.
\end{aligned}
$$

The last line follows from Massart's Finite Lemma (Massart [2000], Lemma 5.2). $\square$

**Theorem 4** (Structural Results for Rademacher Complexity, Theorem 12 of Bartlett and Mendelson [2002])**.** *Let $\mathcal{F}, \mathcal{F}_1, ..., \mathcal{F}_k$ and $\mathcal{H}$ be classes of functions. Then,*

1. *If $\mathcal{F} \subseteq \mathcal{H}$, $\mathfrak{R}_n(\mathcal{F}) \le \mathfrak{R}_n(\mathcal{H})$.*

2. *$\mathfrak{R}_n(\sum_{i=1}^{k} \mathcal{F}_i) \le \sum_{i=1}^{k} \mathfrak{R}_n(\mathcal{F}_i)$.*

3. *For any $L$-Lipschitz function $h$, $\mathfrak{R}_n(\mathcal{F} \odot h) \le 2L \cdot \mathfrak{R}_n(\mathcal{F})$.*

# B   Related Work

**Post-hoc explainability**. In contrast to designing fully-interpretable models such as GAMs (Generalized Additive Models), a dominant line of work in interpretable machine learning is of *post-hoc* explainability. *Post-hoc* methods refers to delivering explanations of model predictions after the prediction has been made. The most popular line of work in this domain is that of *instance-based* explanations, i.e., explaining each prediction via a local input-specific explanation. For example, the work of Ribeiro et al. [2016] introduces LIME (Local Interpretable Model Explanations) that fit a weighted linear regression model, for any input $x$, using random samples generated from $\mathcal{B}_2(x)$, where the weights are computed via the distance of the random point from the true sample. SHAP [Lundberg and Lee, 2017] computes a similar linear explanation based on Shapley values using influence functions derived from cooperative game theory. It has been shown to unify several prior approaches, e.g., DeepLIFT [Shrikumar et al., 2017, 2016], LIME [Ribeiro et al., 2016], and Layerwise Relevance Propagation (LRP) Bach et al. [2015]. Several follow-up works have addressed shortcomings in this line of work, please see Madsen et al. [2021], Chakraborty et al. [2017], Du et al. [2019], Carvalho et al. [2019] for excellent surveys on recent developments in this area. More importantly, however, these approaches are not effective for high-stakes decision-making. They are notoriously unstable [Ghorbani et al., 2019a, Lakkaraju and Bastani, 2020], expensive to compute [Slack et al., 2021], unjustified [Laugel et al., 2019] and in many cases, inaccurate [Lipton, 2018]. Rudin [2019] outlines several of these shortcomings in incredible detail, and hence makes the case to replace *post-hoc* interpretable approaches with methods that are inherently explainable.

**Transparent and Interpretable Machine Learning**. Early work has focused on greedy or ensemble approaches to modeling interactions [Friedman, 2001, Friedman and Popescu, 2008] that enumerate pairwise interactions and learn additive interaction effects. Such approaches often pick up spurious interactions when data is sparse [Lou et al., 2013] and are impossible to scale to modern-sized datasets due to enumeration of individual combinations. As an improvement, Lou et al. [2013] proposed $GA^2M$ that uses a statistical test to filter out only "true" interactions. However $GA^2M$ fails to scale to large datasets as it requires constant re-training of the model and ad-hoc operations such as discretization of features which may require tuning for datasets with a large dimensionality. Other generalized additive models require expensive training of numerous decision trees, kernel machines or splines [Hastie and Tibshirani, 2017], which make them unattractive compared to black-box models.

An alternate approach is is to learn interpretable neural network transformations. Neural Additive Models (NAMs, Agarwal et al. [2021]) learn a DNN per feature. TabNet [Arık and Pfister, 2021] and NIT [Tsang et al., 2018] alternatively modify NN architectures to increase their interpretability. NODE-GAM [Chang et al., 2021] improves NAMs with oblivious decision trees for better performance while maintaining interpretability. Our approach is notably distinct from these prior works: we do not require iterative re-training; we can learn *all* pairwise interactions regardless of dimensionality; we can train SPAM via backpropagation; and we scale effortlessly to very large-scale datasets.

**Learning Polynomials**. The idea of decomposing polynomials was of interest prior to the deep learning era. Specifically, the work of Ivakhnenko [1971], Oh et al. [2003], Shin and Ghosh [1991] study learning neural networks with polynomial interactions, also known as *ridge polynomial networks* (RPNs). However, RPNs are typically not interpretable: they learn interactions of a very high order, and include non-linear transformation. Similar rank decompositions have been studied in the context of matrix completion [Recht, 2011], and are also a subject of interest in tensor decompositions [Nie, 2017a, Brachat et al., 2010], where, contrary to our work, the objective is to decompose existing tensors rather than directly learn decompositions from gradient descent. Recently, Chrysos et al. [2019, 2020] use tensor decompositions to learn higher-order polynomial relationships in intermediate layers of generative models. However, their work uses a recursive formulation and learns high-degree polynomials directly from uninterpretable input data (e.g., images), and hence is non-interpretable.

# C  Experimental Details

All our benchmarks are implemented in PyTorch and run on a cluster, where each machine was equipped with $8\times$V100 NVIDIA GPU machines with 32GB VRAM. We set the batch size of at most 1024 per GPU, and in case a model is too large to fit a batch size of 1024, we perform a binary search to find the largest batch size that it will accommodate. All datasets have features renormalized between the range $[0, 1]$ for uniformity and simplicity. For all experiments we use the Adam with decoupled weight decay (AdamW) optimizer [Loshchilov and Hutter, 2017].

## C.1  Tabular Dataset Details

We use the following tabular datasets.

1. **California Housing** (CH, Pace and Barry [1997]): This dataset was derived from the 1990 U.S. census, and is a regression task, fitting the median house value for California districts using demographic information. The dataset contains 20,640 instances and 8 numeric features.
2. **FICO HELOC** (FICO, FICO [2018]): This is a binary task, part of the FICO Explainability challenge. The target variable denotes the "risk" of home equity line of credit (HELOC) applicants and features correspond to credit report information. It has 10,459 instances and 23 features.
3. **Cover Type** (CovType, Blackard and Dean [1999]): This multi-class dataset is part of the UCI ML repository [Asuncion and Newman, 2007]. The task is classification of forest cover type from cartographic variables. It contains 581,012 instances and 54 attributes over 6 classes.
4. **20 Newsgroups** (Newsgroups, Lang [1995]): This is a popular benchmark dataset where the problem is multi-class classification of email text into 20 subject categories. Following standard convention, we extract TF-IDF features on the training set, which gives us 18,828 instances and 146,016 features over 20 classes (see Appendix Section C.2 for details).

## C.2  20 Newsgroups Feature Extraction

We use the following feature extraction pipeline from the `sklearn` repository. We first use the `TfIdfVectorizer` to compute the TF-IDF features on only the training set. Next, we transform the testing and validation sets using the learnt vectorizer, and rescale all the features within the range $[0, 1]$. This gives us 18,828 instances and 146,016 features over 20 classes.

## C.3  Concept Bottleneck Implementation

In an effort to scale interpretable approaches to problems beyond tabular datasets, we consider benchmark problems using the "Sequential Concept Bottleneck" framework of Koh et al. [2020]. The general idea of concept bottleneck (CB) models are to enable feature extraction necessary for problems in domains such as computer vision by learning a black-box model that can first predict human interpretable "concepts", and then learn a simple, typically fully-transparent model on these concepts to predict the final category. For example, in the problem of bird species classification from images [Wah et al., 2011], one can consider visible parts and attributes of the birds as intermediary "concepts" on which one can learn a linear classifier in order to provide reasonable classification performance while at the same time providing interpretable decisions. Similar approaches have also been explored in the work of, e.g., Chen et al. [2019], Ghorbani et al. [2019b] and Zhang et al. [2020]. We consider concept bottleneck models for computer vision tasks.

### C.3.1  CUB-200 Concept Bottleneck

The Caltech-UCSD Birds (CUB)-200 [Wah et al., 2011] dataset is an image classification dataset that has images belonging to 200 different species of birds, a problem that is an instance of the broader problem of *fine-grained visual classification*. The dataset is equipped with keypoint annotations of 15 bird parts, e.g., crown, wings, etc., and each part has $\geq 1$ part-attribute labels, e.g., *brown leg*, *buff neck*, etc. For the concept bottleneck model, we train a convolutional neural network architecture that predicts these intermediary concepts from the images, and then used this trained CNN model (with weights fixed) to extract features that are then trained using one of the several benchmark approaches (including SPAM). One key difference between prior implementations and ours in this concept bottleneck setting is that we ignore laterality within the annotations, e.g., "left wing" and "right wing" are treated simply as "wing".

**CNN architecture**. We train a ResNet-50 [He et al., 2016] on images resized to the size $448 \times 448$ resolution until the last pooling layer of the network (`pool5`). The features are 2048-dimensional but ignoring the spatial pooling, are evaluated on each $14 \times 14$ patch within the image. We train part-attribute linear classifiers on these spatially-arranged extracted features, where the part locations are used from the training annotations. Once the network has been trained, we run max-pooling over the $14 \times 14$ grid to extract features that are used in the second stage training. Note that this setup is different from the original Concept Bottleneck [Koh et al., 2020] architecture, as the original setup predicted the keypoints from the entire image, instead of a spatially-supervised version. We use this architecture as the original setup overfit very easily.

### C.3.2 Common Objects Dataset Details

We construct a dataset by collecting public images from Instagram[1], involving common household objects (e.g., bed, stove, saucer, tables, etc.) with their bounding box annotations. For each bounding box, we collect 2,618 interpretable annotations, namely, parts (e.g., leg, handle, top, etc.), attributes (e.g., colors, textures, shapes, etc.), and part-attribute compositions. The dataset has 2,645,488 training and 58,525 validation samples across 115 classes with 2,618 interpretable concepts.

### C.3.3 Hyperparameters

Gradient-descent based methods are trained with the Adam with decoupled weight decay (AdamW) optimizer Loshchilov and Hutter [2017]. All methods are trained for 1000 epochs on the California Housing and FICO HELOC datasets, 100 epochs on iNaturalist Birds and Common Objects datasets, and 500 epochs on the CoverType, Newsgroups and CUB datasets. We use a cosine annealing learning rate across all gradient-descent based approaches and do a random hyperparameter search within the following specified grids.

**Linear (Order 1 and Order 2).** The learning rate range is $[1e{-}5, 100]$, and the weight decay interval is $[1e{-}13, 1.0)$. We use the same interval for $L_1-$regularized models as well.

**Deep Neural Networks.** We experiment with 3 neural network architectures that are all comprised of successive fully-connected layers. The first has 5 hidden layers of the shapes $[128, 128, 64, 64, 64]$ units; the second is three hidden layers deep with $[1024, 512, 512]$ units, and finally we have a shallow but wide architecture having 2 hidden layers with $[2048, 1024]$ units. We report the best accuracies achieved across all three architectures. Moreover, we observed that increasing the depth with more layers did not improve accuracy. We search the initial learning rate in the interval $[10^{-6}, 10.0)$, weight decay in $[1e{-}9, 1.0)$, dropout in the set $\{0, 0.05, 0.1, 0.2, 0.3, 0.4, 0.5, 0.6, 0.7, 0.8, 0.9\}$.

**Neural Additive Models.** We proceed with the standard proposed architectures in the original implementation of Agarwal et al. [2021]. We experiment with two architectures: the first is an MLP with 3 hidden layers and $[64, 64, 32]$ units, and the second is an MLP with 1 hidden layer with $1,024$ units and an ExU activation. We tune the learning rate in the interval $[1e{-}5, 1.0)$, weight decay in $[1e{-}9, 1.0)$, output penalty weight in $[1e{-}6, 100)$, dropout and feature dropout in the set $\{0, 0.05, 0.1, 0.2, 0.3, 0.4, 0.5, 0.6, 0.7, 0.8, 0.9\}$.

**EBMs.** We describe the hyperparameter followed by the range of values we search from.

- Maximum bins: $\{8, 16, 32, 64, 128, 256, 512\}$
- Number of interactions: $\{0, 2, 4, 8, 16, 32, 64, 128, 256, 512\}$ (0 for EBMs, $\geq 0$ for EB$^2$Ms)
- learning rate: $[1e{-}6, 100)$
- maximum rounds: $\{1000, 2000, 4000, 8000, 16000\}$
- minimum samples in a leaf node: $\{1, 2, 4, 8, 10, 15, 20, 25, 50\}$
- maximum leaves: $\{1, 2, 4, 8, 10, 15, 20, 25, 50\}$
- binning: $\{$"quantile", "uniform", "quantile_humanized"$\}$
- inner/outer bags: $\{1, 2, 4, 8, 16, 32, 64, 128\}$.

---

[1] www.instagram.com

**XGBoost.** We describe the hyperparameter followed by the range of values we search from.

- number of estimators: $\{1, 2, 4, 8, 10, 20, 50, 100, 200, 250, 500, 1000\}$
- max-depth: $\{\infty, 2, 5, 10, 20, 25, 50, 100, 2000\}$
- $\eta$: $[0.0, 1.0)$
- `subsample`: $[0.0, 1.0)$
- `colsample_bytree`: $[0.0, 1.0)$

**CART.** We describe the hyperparameter followed by the range of values we search from.

- criterion: {"absolute error", "friedman mse", "squared error", "poisson"}
- splitter: {"best", "random"}
- minimum samples leaf: $\{4, 8, 16, 32, 64, 128, 256\}$
- minimum samples split: $\{4, 8, 16, 32, 64, 128, 256\}$

**SPAM-Linear** . For SPAM-Linear, there are 4 sets of hyperparameters: initial learning rate (tuned logarithmically from the set $[10^{-7}, 10^2]$), weight decay (tuned logarithmically from the set $[10^{-13}, 10^1]$), dropout for the singular values $\lambda$, which is tuned from the range $[0, 1]$, and the vector of ranks $r$ which consequently specifies the degree of the polynomial. For degree 2 SPAM, we search ranks from the set $\mathcal{S} = \{[25], [50], [100], [200], [250], [400], [500], [750], [800], [1000], [1200], [1400], [1600]\}$ and we for order 3, we search from the set $\mathcal{S} \times \mathcal{S}$ (i.e., product set of original range).

**SPAM-Neural** . For SPAM-Neural, in addition to the 4 original hyperparameters, we search from the NAM space identical to the range employed for NAM (please see above). Regarding the original hyperparameters, we search: initial learning rate (tuned logarithmically from the set $[10^{-7}, 10^2]$), weight decay (tuned logarithmically from the set $[10^{-13}, 10^1]$), dropout for the singular values $\lambda$, which is tuned from the range $[0, 1]$, and the vector of ranks $r$ which consequently specifies the degree of the polynomial. For degree 2 SPAM, we search ranks from the set $\mathcal{S} = \{[25], [50], [100], [200], [250], [400], [500], [750], [800], [1000], [1200], [1400], [1600]\}$ and we for order 3, we search from the set $\mathcal{S} \times \mathcal{S}$ (i.e., product set of original range).

# D   Human Subject Evaluation Details

## D.1   CUB-200 Dataset Details

[Figure]

Figure 3: Random sample Images from the CUB-200 dataset used in Human Subject Evaluations
Figure 3 denotes a random sample of images that were shown to the participants. We randomly
sampled images from a pre-selected set of a pair of classes. There were a total of 5 pairs, corresponding
to the following classes:

1. "122.Harris_Sparrow", "060.Glaucous_winged_Gull"
2. "057.Rose_breasted_Grosbeak", "034.Gray_crowned_Rosy_Finch"
3. "066.Western_Gull", "035.Purple_Finch"
4. "064.Ring_billed_Gull", "008.Rhinoceros_Auklet"
5. "095.Baltimore_Oriole", "086.Pacific_Loon"

All images were sampled from the validation set.

## D.2   Experiment Interface

The experiment was implemented using the `Mephisto` [Miller et al., 2017] library in `react.js`.
Please see the subsequent pages for screenshots of the exact interface.

This is a simple experiment on the interpretability of machine learning models.

You will be first shown two sets of 4 images belonging to two different species of birds **Species A** and **Species B**. For each image, you will be shown the species a machine learning model predicted and an explanation of which attributes about the birds the model used to make its prediction.

The explanation is in the form of a list of attributes, along with the importance (positive or negative) the model assigned to each attribute. Your task is to first familiarize yourself with the samples shown, with the objective of understanding which attributes the model uses to make its prediction.

After the inital training phase, you will be shown only the explanations, and you will have to judge which species (between **A** and **B**) the model is likely to have predicted.

You may return back to the training phase at any time to revise the attributes and their explanations for each example.

In the main task, you will be shown a total of 5 images. We recommend you to spend at least 20 seconds for each image.

[ Move To Training Phase ]

Figure 4: Human Subject Experiment: Introduction page

[Figure]

Figure 5: Human Subject Experiment: Example training phase, class **A** for linear explanations.

[Figure]

Figure 6: Human Subject Experiment: Example training phase, class **B** for linear explanations.

[Figure]

Figure 7: Human Subject Experiment: Example training phase, class **A** for SPAM (order 2) explanations.

[Figure]

Figure 8: Human Subject Experiment: Example testing phase, for linear explanations.

[Figure]

Figure 9: Human Subject Experiment: Example testing phase, for SPAM explanations.