# OpenReview forum: "Scalable Interpretability via Polynomials"
_NeurIPS.cc/2022/Conference — NeurIPS 2022 Accept_

### Official Review · Reviewer_arrH · 2022-06-26

**Rating:** 6
**Confidence:** 4
**Soundness:** 3 good
**Presentation:** 3 good
**Contribution:** 3 good

**Summary:**

This paper proposes a fully-interpretable model which is in the line of generalized additive models (GAMs). In order to let the fully-interpretable model scalable, the authors reduce the conventional GAMs to a polynomial one. By doing tensor decomposition, the proposed model can be effortlessly scalable and model higher-order feature interactions without a combinatorial parameter explosion. Experimental results validate the effectiveness of the proposed method both in the perspectives of prediction and human evaluation. Theoretical results also guarantee their model’s performance.

**Questions:**

What if we continue to increase the length of explanation? Is it the 7 global optimal in Figure 2C?

**Ethics Review Area:**

["I don’t know"]

**Limitations:**

Yes.

**Strengths And Weaknesses:**

Strengths:
1. The technique of this paper is sound. The authors provide a clear view and logic of the proposed idea with the basic form of the polynomial additive models and how it can be scalable by low-rank decompositions. The improving on geometric rescaling and shared bases for multi-class problems also make sense.
2. So is the clarity of this paper according to the above comments.
3. The human subject evaluation is also convincing.

Weaknesses:
1. It would be better to provide the details in the human subject evaluation. For example, the authors could provide the explanation provided by the models in the bird prediction task. Then we may have a better understanding on how the model is interpreted.
2. The significance of the proposed method seems fair. The proposed method is not post-hoc, which means the prediction and interpretation all count on the proposed method. Table 1 shows pretty good prediction results where the proposed method can mostly outperform all the baselines including DNN. But it seems a bit tricky since the DNN is just a simple MLP. Outperforming an MLP may not be a big advantage. In addition, what are the reasons to choose the four datasets presented in the paper? I am concerned that the results are cherry picking so that the proposed method cannot work well in prediction. If so, a slightly better interpretable ability may not be worth applying to real world tasks. Good prediction plus post hoc explanation may be sufficient.

---

> ### Author Response · Authors · 2022-08-01
> **Author Response**
>
> We would like to thank the reviewer for the positive feedback and the great suggestions. We answer the questions below:
>
> - *It would be better to provide the details in the human subject evaluation. For example, the authors could provide the explanation provided by the models in the bird prediction task. Then we may have a better understanding on how the model is interpreted.*
>   - Thanks for the suggestion! We have included some sample explanations on the CUB dataset outputted by MLP+LIME and SPAM Order 2 models in the Appendix Section D along with more details about the human subject evaluation. We will include a short discussion in the main paper as well.
>
> - *“In addition, what are the reasons to choose the four datasets presented in the paper? I am concerned that the results are cherry picking so that the proposed method cannot work well in prediction.”*
>   - Thank you for raising this concern! We would first like to clarify that we evaluate on 7 datasets and not 4. Next, to address cherry-picking: on fact, during our dataset selection procedure, we gave great care to have datasets that satisfied:
>      - (a) gap between linear and non-linear classifiers: several interpretability benchmark datasets have small performance gaps between linear and non-linear classifiers, where it is obvious that for interpretability one should prefer a linear model. Therefore, we wanted to select some datasets where there was in fact a tradeoff between interpretability and performance so that we could highlight the improvements of SPAM-like models.
>     - (b) scale: we wanted to explore all possible dataset scale - large-scale in number of features (e.g., Newsgroups, 150K features), large-scale in number of samples (e.g., Common Objects, 2.6M samples), as well as small scale datasets.
>     - However, to assuage any concerns of dataset selection, we have run additional experiments on 9 extra tabular datasets that are commonly present in the literature. The results of this are summarized in Table 1 in the common response above. We find that on several datasets, order 2 interactions match XGBoost performance. We will include a full version of this comparison with order 3 interactions as well in the main paper.  We will extend our experimental section to include all these new datasets, making our total benchmark 16 datasets, the largest evaluation of interpretable models to the best of our knowledge.
>
> - *“But it seems a bit tricky since the DNN is just a simple MLP. Outperforming an MLP may not be a big advantage.”*
>   - The reviewer is indeed correct that for some datasets MLPs might not be enough. We do, in fact, compare with XGBoost as well for all methods that can support it. We agree that it is indeed possible to extract some benefits on top of XGBoost/MLPs using some specific architectures, however, we believe that is tangential to the objective of this paper, which is to highlight a significant improvement in fully-transparent models when interpretability is a requirement. As our human subject evaluations suggest, current post-hoc explanations are not reliable for very large-scale datasets, and hence the precise model being used is extremely dependent on the application and requirement of interpretability. Nevertheless, this is a valuable point that the reviewer has highlighted and we will be sure to address this in the main paper as well.
>
> - *“What if we continue to increase the length of explanation? Is it the 7 global optimal in Figure 2C?”*
>   - Thanks for this point! Indeed, a short explanation appears to be more faithful and useful from a human subject perspective. We did increase the length of the explanation to 15 and found that the performance decreased further. We believe this is due to excessive information that can be confusing for users. We will add this point in the main paper.

---

> ### Comment · Area_Chair_f2Jq · 2022-08-09
> **Author response phase closing today**
>
> The author-response phase closes today. Please acknowledge the author rebuttal and state if your position has changed. Thanks!

---

### Official Review · Reviewer_othi · 2022-07-10

**Rating:** 7
**Confidence:** 3
**Soundness:** 3 good
**Presentation:** 4 excellent
**Contribution:** 3 good

**Summary:**

This paper proposes SPAM which extends Generalized Additive Models (GAM) with polynomials. SPAM learns a polynomial model that can capture any-order interactions among features. The authors leverages the proved symmetric property and assumed low-rank property of the weight matrices to simplify the optimization problem. For multiple-class classification tasks, the authors propose to use shared bases to further speed up training. The authors show under some assumption, the generalization error of SPAM scales exponentially w.r.t. the chosen low-rank order. The authors conduct experiments with several datasets and show that proposed SPAM achieve better results than various of baseline models. The authors also conduct human evaluation on how good SPAM is comparing with a black box model and the results show SPAM achieve much better results than using a black box model with post-hoc explanation.

**Questions:**

- Can you show some empirical resutls about the effects of using shared bases? I'm curious to see how much that trick would improve. Do you think it makes sense to use a shared bases for $u_1, u_{2i}, ..., u_{ki}$ in equation 3?

**Ethics Review Area:**

["I don’t know"]

**Limitations:**

- I think it's worth noting the running time of the SPAM on different datasets to help understand the scalability of SPAM. I guess for extremely high-dimension datasets (like high-resolution images or videos or texts), SPAM might not be able to train efficiently. It would be great if the authors can provide some discussions on this part.

**Strengths And Weaknesses:**

Strengths:
- The idea of using polynomials to improve GAM is novel. The authors address the challenges of using polynomials well (mainly, scalability) by leveraging some nice properties of the problem.
- The authors provide thorough analysis for the proposed models. The charaterization of the generalization error, the geometric rescaling, data preprocessing, and shared bases idea for multi-class problems help the readers get a deeper understanding of the proposed model.
- The experimental section is thorough. Other than regular comparsion with baseline methods, the authors check the effects of order, sparsity assumtion for higher order interactions and verify the generalization error empirically. The human evaluation setting is novel and provide more evidence on why a fully-inpretably model might be better.

Weaknesses:
- For Table 1, I think the authors can include results for NA^2M since SPAM at lease uses order-2 interactions.

---

> ### Author Response · Authors · 2022-08-01
> **Author Response**
>
> We would like to thank the reviewer for their assessment of the paper and their suggestions. Please find responses to your questions below:
>  - *Results for $NA^2M$*:
>    - Thank you for pointing out that issue! Please see Table 2 in the common response above for the comparison of $NA^2M$ with SPAM and NeuralSPAM (Order 2). We can see that while SPAM by itself doesn’t always beat $NA^2M$ (due to the lack of nonlinearities), NeuralSPAM outperforms $NA^2M$ on all but 1 dataset. On CoverType, we believe that NeuralSPAM has too few parameters compared to $NA^2M$ (which has one MLP for each possible combination) and hence underfits severely. When we increase the number of subnets to 8 for NeuralSPAM we obtain an accuracy of 0.9022 on Order 2, whereas it is impossible to train with 8 subnets for $NA^2M$ for order 2 due to memory usage. Moreover, adding subnets does not improve $NA^2M$ performance significantly.
>
> - *Empirical Results for Shared Bases:*
>   - Please see Table 3 in the common response above for the change for multi-class problems. We observe a consistent improvement from sharing bases on all datasets due to the reduction in the number of parameters. We will include this in the final paper, thank you for the suggestion!
>
> - *Shared bases across different degrees*:
>   - That is an excellent suggestion! While we haven’t explored that in our current set of experiments, we will update with a comparison with sharing bases across degrees in the final version.
>
> - *Scalability of SPAM*:
>   - Thanks for this point! In practice, since the SPAM dimensionality increases only logarithmically with the dimensionality of the data, Linear-SPAM can be scaled without issue to datasets with 150K features (as in Newsgroups), in contrast to other interpretable approaches like NAM or EBMs.
>   - We agree that Neural-SPAM will have less scalability due to the non-linear mapping. However, if we contrast that growth with previous work such as NAM, our growth is still $\mathcal O(dk)$ for a $d$-dimensional input with degree $k$ interactions, whereas NAM, for instance, scales as $\mathcal O(d^k)$ for degree $k$ interactions.
>   - Regarding images and structured data: We expect that SPAM will be applied on interpretable concepts for images/text, e.g., as done via concept bottleneck models. One can, however, apply SPAM to the output of a black-box model as well without issue, e.g., ResNet-50 output features, to improve performance.
>
> - *Running time of SPAM*:
>   - Thanks for this suggestion! Please see Table 4 in the common response above for runtime comparison of LinearSPAM with NAM on some datasets. We will include this discussion in the main paper as well. The summary of this comparison is that Linear versions of SPAM perform significantly faster than NAM and MLPs, even at order  3. For NeuralSPAM, we observe that it is faster than NAMs (order 2), however the additional parameters indeed decrease the throughput.

---

> ### Comment · Area_Chair_f2Jq · 2022-08-09
> **Author rebuttal phase closing today**
>
> The author-rebuttal phase closes today. Please acknowledge the author rebuttal and state if your position has changed. Thanks!

---

### Official Review · Reviewer_wzTc · 2022-07-11

**Rating:** 7
**Confidence:** 4
**Soundness:** 3 good
**Presentation:** 4 excellent
**Contribution:** 3 good

**Summary:**

The paper proposes to make Generalized Additive Models (GAMs) scalable, by tensor rank decompositions of polynomials. Specifically, the traditional GAMs are re-written into the tensor computation form. The weight matrices are then processed with rank decomposition, making the GAMs a series of inner products and are more computationally efficient. The authors also propose several tricks to help learning, including feature rescaling, sharing basis across classes, and extending GAMs with non-linear operations. Experiments under different data domains are conducted, including MLPs and CNNs. The experiments make use of the concept bottleneck backbone to further improve its performances. Human subject evaluations are also included since this paper is related with interpretability. This is a well-written paper, with in-depth understanding of GAMs and comprehensive experiments.

**Questions:**

Please see the Weakness part.

**Limitations:**

The paper proposes a series of extensions of SPAM to address some possible limitations.

**Strengths And Weaknesses:**

Strengths
1. The paper addresses an important problem in interpretable machine learning, i.e., how to build efficient and effective models besides interpretability. The paper chooses GAMs as prototypes, making a smart application of tensor decomposition to make GAMs more efficient. The paper is very well-written.
2. The paper considers a series of extensions for the proposed SPAM models. These extensions are consistent with the core contribution.
3. Comprehensive experiments are conducted. With both quantitative analysis and human studies.

Weaknesses
1. I would avoid adding "fully-interpretable models" in the abstract. Is SPAM really "fully-interpretable"? I think "inherently interpretable" could be better.
2. I think adding some more analysis about the learned SPAM models (i.e., which features or feature interactions are important) could help readers understanding what has been learned inside models for some of the tasks.

---

> ### Author Response · Authors · 2022-08-01
> **Author Response**
>
> We would like to sincerely thank the reviewer for their appraisal of our paper. Please find responses to your concerns below:
>
> - *I would avoid adding "fully-interpretable models" in the abstract. Is SPAM really "fully-interpretable"? I think "inherently interpretable" could be better.*
>
>   - Thank you for the remark! We agree that inherently interpretable is a better characterization of our approach, as fully-interpretable is a stricter definition that polynomials will not obey in certain cases (e.g., large degree of polynomial).
>
> - *I think adding some more analysis about the learned SPAM models (i.e., which features or feature interactions are important) could help readers understanding what has been learned inside models for some of the tasks.*
>
>   - Thank you for the excellent suggestion! In the Appendix Section D we present examples of explanations that we used for the CUB-200 dataset in our human subject evaluations. We will provide, in the main paper, some examples of explanations for different datasets as well.

---

> > ### Comment · Reviewer_wzTc · 2022-08-05
> > **Thanks for the reply**
> >
> > I do not have futher concerns if the above will be addressed in the future version.

---

### Author Response · Authors · 2022-08-01
**Common Response and New Experiments for Rebuttal**

We would like to sincerely thank the reviewers for their positive feedback and highly constructive feedback. We will be sure to incorporate it into our final version.

## Comparison on additional benchmarks

In addition to the 7 datasets we considered in the main paper, we evaluate on 9 additional benchmark datasets common in the tabular learning literature. We consider the MIMIC2, Credit and COMPAS datasets from the Neural Additive Models (NAM) work [Agarwal et al 2021], and the Click, Epsilon, Higgs, Microsoft, Yahoo and Year datasets from [Chang et al, 2021]. The result of this is summarized in Table 1. For MIMIC2, Credit and COMPAS we report the AUC (higher is better). For Epsilon, Higgs and Microsoft, we report error rate (lower is better). For the remaining regression tasks we report MSE (lower is better). The best interpretable model is in **bold**.

### Table 1

| Method               | MIMIC2 (AUC) | Credit (AUC) | COMPAS (AUC) | Click (ERR) | Epsilon (ERR) | Higgs (ERR) | Microsoft (MSE) | Yahoo (MSE) | Year (MSE) |
| -------------------- | ------------- | ------------- | --------------- | -------------- | -------------- | -------------- | ---------------- | -------------- | ----------- |
| NAM                  | 0.8539        | 0.9766        | 0.7368          | 0.3447         | 0.1079         | 0.2972         | 0.5824           | 0.6093         | 85.25       |
| NODE-GAM             | 0.832         | 0.981         | 0.742   | **0.3342** | 0.1040 | 0.2970 | 0.5821   | 0.6101 | 85.09       |
| LinearSPAM (Ord 2) | 0.8514        | 0.9836        | **0.7426**          | 0.3791         | **0.1011**         | 0.2881         | 0.571            | 0.5923         | 81.306      |
| NeuralSPAM (Ord 2) | **0.8664**        | **0.9850**        | 0.7411          | 0.3348         | 0.1020         | **0.2750**        | **0.5671**           | **0.5869**         | **79.99**       |
| XGBoost              | 0.843         | 0.978         | 0.744           | 0.3334         | 0.1112         | 0.2328         | 0.5544           | 0.5420         | 78.53       |



## Comparison with $NA^2M$

We compare NeuralSPAM with $NA^2M$ (NAM with pairwise features) in Table 2. SPAM by itself doesn’t always beat $NA^2M$ (due to the lack of nonlinearities), NeuralSPAM outperforms $NA^2M$ on all but 1 dataset. On CoverType, we believe that NeuralSPAM has too few parameters compared to $NA^2M$ and hence underfits severely. When we increase the number of subnets to 8 for NeuralSPAM we obtain an accuracy of 0.9022 on Order 2, whereas it is impossible to train with 8 subnets for $NA^2M$ for order 2 due to memory usage. For CH dataset, we report MSE (lower is better). For the rest, we report AUC or ACC (higher is better).


### Table 2

| Method               | CH (RMSE) | FICO (AUC) | CovType (AUC) | CUB (ACC) | iNat (ACC) |
| -------------------- | ---------- | ----------- | ---------------- | ------------- | ---------- |
| NAM (Order 2)        | 0.4921     | 0.7992      | 0.8872            | 0.7713     | 0.4591        |
| NeuralSPAM (Order 2) | 0.4914     | 0.8011      | 0.7770       | 0.7762     | 0.4689        |




## Effect of sharing bases in SPAM in multi-class problems
We observe a consistent improvement from sharing bases on all datasets due to the reduction in the number of parameters. We report accuracy (higher is better).

### Table 3

| Method                          | News (ACC) | CUB (ACC) | iNat (ACC) | Comm Obj.(ACC) |
| ------------------------------- | ------------- | ---------- | ------------- | ---------------- |
| LinearSPAM without shared bases | 0.8334        | 0.7575     | 0.4202        | 0.2195           |
| LinearSPAM                      | 0.8472        | 0.7786     | 0.4605        | 0.2361           |




## Runtime comparison of LinearSPAM and NeuralSPAM with NAM, MLP

We report the throughput (in terms of samples per second) for 4 datasets on different models. The summary of this comparison is that Linear versions of SPAM perform significantly faster than NAM and MLPs, even at order 3. For NeuralSPAM, we observe that it is faster than NAMs (order 2), however the additional parameters indeed decrease the throughput.

### Table 4

| Method               | CH       | FICO     | CovType  | News     |
| -------------------- | -------- | -------- | -------- | -------- |
| NAM                  | 5x10^5   | 1.2x10^5 | 8x10^4   | 23       |
| NAM (Order 2)        | 1.1x10^4 | 6000     | 3000     | \-       |
| LinearSPAM (Order 2) | 6.1x10^7 | 6.7x10^7 | 6.1x10^7 | 2.6x10^6 |
| NeuralSPAM (Order 2) | 1.7x10^5 | 7912     | 4103     | \-       |
| LinearSPAM (Order 3) | 3.2x10^7 | 3.7x10^7 | 3.9x10^7 | 1.8x10^5 |
| NeuralSPAM (Order 3) | 1.1x10^5 | 5322     | 2681     | \-       |
| MLP (Small)          | 1.3x10^7 | 1.3x10^7 | 1.3x10^7 | 2.2x10^5 |
| MLP (Big)            | 4.6x10^6 | 5x10^6   | 4.5x10^6 | 5.4x10^4 |

---

### Meta-Review · Area_Chair_f2Jq · 2022-08-28

**Recommendation:** Accept
**Confidence:** Certain

**Metareview:**

The paper notes that polynomial functions are inherently interpretable models, and takes algorithmic advantage of the connection between polynomials and tensors by learning the coefficients of the polynomials using a low-rank tensor factorization. The resulting algorithm is shown to outperform prior SOTA interpretable models and to match blackbox model performance on several data sets.

**Award:**

No

---

### Decision · Program_Chairs · 2022-09-14

Accept